# PerturboLLaVA: Reducing Multimodal Hallucinations with Perturbative Visual Training

**Cong Chen**[1,*], **Mingyu Liu**[1,*], **Chenchen Jing**[3], **Yizhou Zhou**[2],
**Fengyun Rao**[2], **Hao Chen**[1], **Bo Zhang**[1], **Chunhua Shen**[3,1]

[1] Zhejiang University    [2] WeChat Group    [3] Zhejiang University of Technology

## Abstract

This paper aims to address the challenge of hallucinations in Multimodal Large Language Models (MLLMs) particularly for dense image captioning tasks. To address the challenge, we identify the current lack of a metric that finely measures the quality of the caption at the concept level. We hereby introduce HalFscore, a novel metric built upon the language graph that is designed to evaluate both the accuracy and completeness of dense captions at a granular level. Additionally, we identify the root cause of hallucination as the model's over-reliance on its language prior. To address this, we propose PerturboLLaVA, which reduces the model's reliance on the language prior by incorporating adversarially perturbed text during training. This method enhances the model's focus on visual inputs, effectively reducing hallucinations and producing accurate, image-grounded descriptions without incurring additional computational overhead. PerturboLLaVA significantly improves the fidelity of generated captions, outperforming existing approaches in handling multimodal hallucinations and achieving improved performance across general multimodal benchmarks.

## 1 Introduction

Multimodal large language models (MLLMs) (Liu et al., 2024b; Zhu et al., 2023; Dai et al., 2023; Bai et al., 2023; Zhang et al., 2023; Lu et al., 2024a; Zhou et al., 2024) have achieved significant strides in complex visual tasks by integrating the world knowledge and reasoning capabilities of large language models (LLMs). Nonetheless, hallucination (Liu et al., 2023a; Zhou et al., 2023; Yin et al., 2023; Wang et al., 2023a) issue persists in these models, and even the most capable multimodal models often respond with texts that do not accurately reflect the provided visual content. Notably, scaling up model parameters and training data has not proven effective inmitigating this issue for MLLMs, unlike for their unimodal language counterparts.

In this paper, we focus on addressing the problem of hallucination in the context of dense image captioning (Liu et al., 2024b; Chen et al., 2023; 2024), which requires a comprehensive and detailed description of every aspect of an image. Dense image captioning exposes hallucinations more acutely, as models must generate rich and detailed captions for complex scenes while ensuring fidelity to the visual content. Hallucinations undermine the reliability of MLLMs in applications that require precise and faithful visual descriptions. To tackle the hallucination challenge in MLLMs, we recognize the need for a robust quantitative metric that accurately reflects caption quality regarding hallucination. Hence, we introduce *HalFscore*, a novel metric providing a more granular and comprehensive evaluation of hallucinations specific to dense captioning. HalFscore measures both the accuracy and completeness of dense captions by identifying the incorrect elements and assessing the missing details, offering a balanced view of the model's performance. To achieve this, we propose to build the language graph that captures the main concepts along with their relationships, and compute its discrepancy against the ground truth. HalFscore aggregates the precision and recall to model the accuracy and completeness of dense captioning results. Compared to previous hallucination scores (Li et al., 2023b; Guan et al., 2024), the proposed HalFscore offers a more fine-grained and holistic evaluation, serving as a valuable guide when developing the hallucination suppression method. We further analyze the common cases plagued with hallucinations, and conjecture that the issue in MLLMs comes from the over-reliance on the model's pretrained linguistic knowledge. As shown

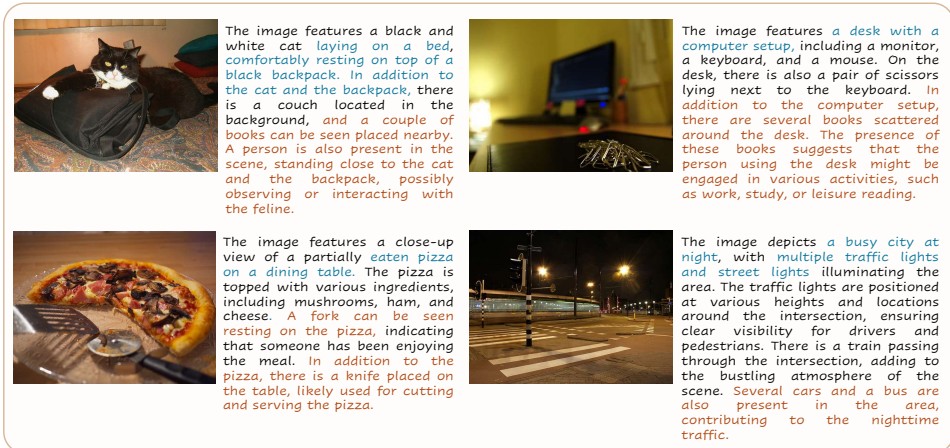

Figure 1: The multimodal model is prone to hallucinate text due to the inherit language bias. Here, the hallucinated text is induced by the preceding concepts generated by the language model.

in Figure 1, as more texts are produced, the multimodal model gradually deviates from multimodal generation to unimodal mode, which favors text generation based on the preceding language patterns while overlooking the actual visual information. For example, when presented with an image of a green banana, a model has the tendency to describe it as yellow because of the common knowledge that ripe bananas are typically yellow. In fact, concurrent MLLMs are obtained by continuous training from a pre-trained LLM and are generally equipped with a strong language bias.

Inspired by this, we propose a simple and effective training strategy that reduces the model's heavy dependence on the language prior by incorporating adversarially perturbed text during training. Specifically, we introduce carefully designed perturbations that aligns with the general knowledge but conflict with the visual content, intentionally misleading the model based on its language prior. For example, we might introduce the perturbation "As bananas ripen, their color gradually turns yellow" before asking about the color of a green banana in the image. This perturbative training enforces the model to scrutinize the image content when predicting every token, rather than hallucinating contents from the text hints. Essentially, our method adjusts the model's conditional distribution to depend more heavily on the image and less on the perturbation text, which leads to more robust multimodal capability.

As opposed to state-of-the-art methods that resort to more advanced decoding strategies (Leng et al., 2024; Huang et al., 2024), the proposed method, *PertuboLLaVA*, effectively suppresses the hallucinations in MLLMs without incurring additional training or inference costs, making it more suitable for real-world applications. Figure 17 shows our method can describe rich image details with less hallucinations. On the other hand, our method is much more efficient, scalable, and easier to adopt compared to RLHF-based methods which require additional human preference data and incur substantial training overhead and complexity. Additionally, we find the proposed method beneficial to general multimodal abilities, bringing boosted performance across all the multimodal benchmarks.

To summarize, our contributions are two-fold. First, we introduce a more principled metric computed on the language graph, serving as a comprehensive hallucination measure. Second, we identify the root cause of hallucinations in MLLMs as its inherent language bias, and propose perturbative visual training, enhancing the model's focus on visual content during training. The proposed method integrates seamlessly into existing training pipelines, introducing minimal additional cost. It provides a scalable, efficient solution to enhance multimodal models' visual understanding capabilities, excelling over prior compared to state-of-the-art methods across multiple dimensions.

## 2 RELATED WORK

### 2.1 MULTIMODAL FOUNDATIONAL MODEL

With the rapid advancement of large language models (Touvron et al., 2023a;b; Chiang et al., 2023; Bi et al., 2024), researchers are leveraging their knowledge and reasoning abilities to build multi-

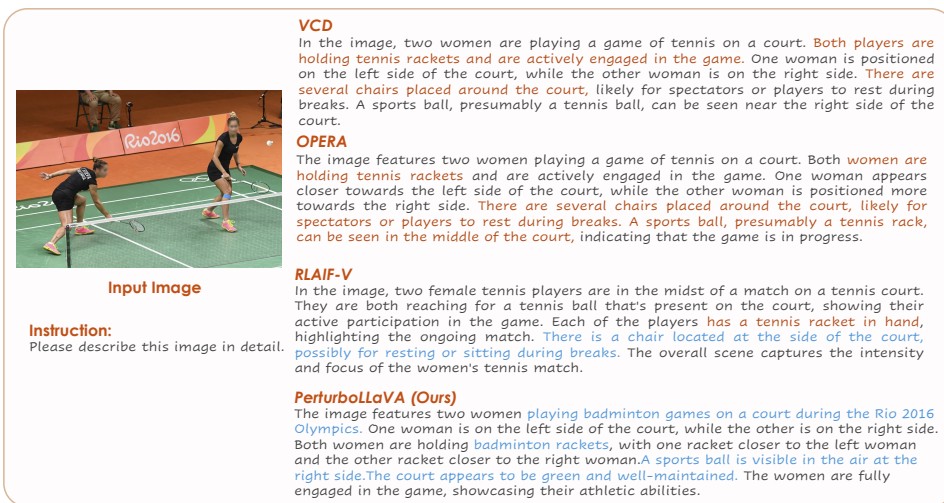

Figure 2: Comparison against the state of the art methods. Hallucinations are highlighted in red, whereas the image detailed descriptions are shown in blue. The proposed PerturboLLaVA describes rich image details more accurately.

modal systems for complex tasks. These models are typically constructed with a pretrained vision encoder to process visual information, a language model backbone responsible for reasoning, and a projector to map the visual data into textual space. The training process of multimodal model is generally divided into two stages: pretraining, using image captioning data for modality alignment, and instruction fine-tuning, using question-answer data to enable task handling. Several leading open-source projects (Bai et al., 2023; Lu et al., 2024b; OpenGVLab, 2024) have amassed large multi-modal datasets and developed high-performing models using advanced language model backbones and vision encoders. However, hallucinations remain a persistent challenge in models' outputs.

## 2.2 EVALUATION OF HALLUCINATION

Various benchmarks assess hallucination in MLLMs, divided into, categorized into close-ended (Li et al., 2023b; Wang et al., 2023b) and open-ended tasks (Sun et al., 2023; Liu et al., 2023b). Close-ended tasks use yes-or-no or multiple-choice questions to test for hallucinations, focusing on accuracy. The POPE(Li et al., 2023b) benchmark detects non-existent entities, while AMBER Wang et al. (2023b) also considers attributes and relationships. In open-ended tasks, such as image captioning or free-form Visual Question Answering (VQA) (Wu et al., 2017; Jing et al., 2020), LLM-free metrics like CHAIR (Rohrbach et al., 2018) measure the ratio of hallucinated to actual objects in responses. On the other hand, LLM-based metrics, such as MMHalBench (Sun et al., 2023) and GAVIE (Liu et al., 2023b), utilize external LLMs like GPT (Achiam et al., 2023) to assign scores to generated responses and are used in benchmarks. Hallucination evaluation in multi-modal models is more evident in open-ended tasks, as these tasks require a detailed understanding of the image and dense outputs. However, current metrics like object-level CHAIR and caption-level MMHalBench lack fine-grained hallucination analysis.

## 2.3 MITIGATION OF HALLUCINATION

Current efforts to mitigate hallucinations are mainly divided into training-based and decoding-based strategies (Leng et al., 2024; Huang et al., 2024). Mainstream training-related approaches (Yu et al., 2024; Sun et al., 2023) introduce algorithms like RLHF (Ouyang et al., 2022) and DPO (Rafailov et al., 2024) from the LLM area into multimodal models. By constructing hallucination preference data, they train a reward model to provide reward supervision or use DPO to reduce multimodal hallucinations. However, the primary challenge with training-based approaches lies in the substantial computational overhead, as they necessitate training additional reward models or incorporating extra training phases. Decoding strategies include approaches like OPERA (Huang et al., 2024),

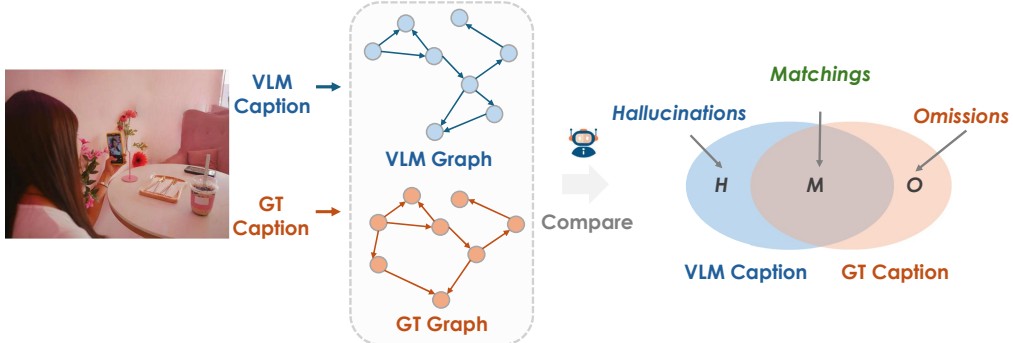

Figure 3: The diagram of computing HalFscore. We construct the language graph to model both the concepts and their relationships for captions. We can then compare the graphs and identify the hallucinations, omissions and matchings between the two sets of concepts respectively.

which corrects abnormal attention map, and methods like VCD (Leng et al., 2024) that decouple language priors causing hallucinations, subtracting them from prediction probabilities. Although decoding strategies have the advantage of being training-free, they do not address the root cause of hallucinations in multimodal models, as these issues originate during training. Moreover, from a practical standpoint, the inference cost for large models often exceeds the training cost, since models are trained once but deployed countless times. In the work, we propose a novel, simple yet effective training strategy that avoids the additional training overhead of previous training-based approaches while offering a more comprehensive solution.

## 3 HUALLUCINATION F-SCORE

An effective hallucination metric should be both fine-grained and comprehensive. Current metrics fall short of these criteria. For example, CHAIR focuses on matching objects while failing to measure hallucinations about attributes and relationships. MMHalbench uses GPT-4 to produce a single holistic score but lacks detailed analysis. Moreover, prior metrics only measure the degree of hallucination without assessing the comprehensiveness of the image captioning, typically favoring a short but confident answer, which is inconsistent with users' subjective experience.

**A Fine-Grained & Comprehensive Hallucination-Metric** We introduce HalFscore which measures both hallucination and completeness of dense captions with fine granularity. As illustrated in Figure 3, we construct graphs for both the MLLM's output and the ground truth. Here, we leverage dense annotations as ground truth, which provides sufficient detailed descriptions that reflect all the aspects of the input images. Specifically, we selected 1,000 images from the Densely Captioned Images (DCI) dataset (Urbanek et al., 2023), in which images are manually annotated and densely captioned. By comparing the graphs, we can identify hallucinations—concepts generated by the model that contradict the ground truth, and omissions—concepts present in the ground truth but absent in the model's captions. We denote the concepts denoted in the generation as $\mathcal{C}_{\text{gen}}$, the concepts corresponding to ground truths as $\mathcal{C}_{\text{gt}}$, and compute the precision and recall as

$$\text{Precision} = \frac{|\mathcal{C}_{\text{gen}} \cap \mathcal{C}_{\text{gt}}|}{|\mathcal{C}_{\text{gen}}|} = 1 - \frac{|\mathcal{C}_{\text{hallucinated}}|}{|\mathcal{C}_{\text{gen}}|}, \tag{1}$$

$$\text{Recall} = \frac{|\mathcal{C}_{\text{gen}} \cap \mathcal{C}_{\text{gt}}|}{|\mathcal{C}_{\text{gt}}|} = 1 - \frac{|\mathcal{C}_{\text{omitted}}|}{|\mathcal{C}_{\text{gt}}|}. \tag{2}$$

Here, the precision reflects the hallucination degree, whereas the recall assesses how well the captions cover image details. Then we compute HalFscore by aggregating these two scores, serving as a single metric that reflects the overall captioning quality:

$$\text{HalFscore} = 2 \times \frac{\text{Precision} \times \text{Recall}}{\text{Precision} + \text{Recall}}. \tag{3}$$

**Graph Computation**  To derive the above HalFscore, graph construction and matching are pivotal. We propose a novel triplet data structure to represent the information for captions. Specifically, we prompt the GPT-4o model to parse the concepts along with their relative relationships, and represent them by the triplet representation, *i.e.*, $\langle e_1, e_2, r_{12} \rangle$. As shown in Figure 4, the triplet representation can well model general and rich information, such as relative relationship between instances, *e.g.*, $\langle \text{clock}, \text{on}, \text{wall} \rangle$, or the attribute description for a single instance, *e.g.*, $\langle \text{mirror}, \text{is}, \text{pink} \rangle$. In this way, the caption is transformed to a set of triplets, serving as a structured representation.

Figure 4: Graph construction. We extract triplets from the caption and build the graph accordingly.

Meanwhile, the triplet can be viewed as two nodes representing entities, $e_1$ and $e_2$, with a relation edge $r$. We then organize the extracted triplets into a concept graph $\mathcal{G}$ by matching entity nodes. In this graph, nodes represent entity concepts $e_i$, and edges represent relational concepts $r_{ij}$ between entities $e_i$ and $e_j$. This graph allows us to represent all the information in the caption comprehensively and accurately. Formally, we obtain the graph $\mathcal{G}$ with nodes $\mathcal{V}$ and edges $\mathcal{E}$ as follows:

$$\mathcal{G} = (\mathcal{V}, \mathcal{E}), \quad \mathcal{V} = \{e_1, e_2, \ldots, e_n\}, \quad \mathcal{E} = \{r_{12}, r_{23}, \ldots, r_{mn}\}. \tag{4}$$

By matching the constructed graphs, we can identify the hallucinated concepts ($\mathcal{C}_{\text{hallucinated}}$) and omitted concepts ($\mathcal{C}_{\text{omitted}}$) using GPT-4o. Based on these identified concepts, we proceed to calculate precision and recall according to the equations aforementioned, and then derive the Fscore. Please see the Appendix A.8 for detailed GPT-4o prompts used for triplet extraction and graph matching.

# 4  MITIGATION OF HALLUCINATION VIA PERTURBATIVE VISUAL TRAINING

## 4.1  PERTURBATIVE VISUAL TRAINING

To mitigate the over-reliance on language priors in multimodal models, we introduce a novel training framework that introduces adaptive, context-specific perturbations in the textual inputs during training. This approach simulates the effect of language priors and forces the model to adjust its responses based on visual data rather than textual biases.

Specifically, during the instruction tuning training, the multimodal model is tasked with predicting the text according to the input image-question tokens $(I, x_q)$. We introduce a contextually adaptive perturbation text $x_p$, crafted to mimic misleading language priors. Thus, we obtain the perturbed input tokens as $(I, x_p, x_q)$, as depicted in Figure 5. This perturbation is integrated seamlessly as part of the input, without any direct loss computation on $x_p$ itself, thus maintaining the integrity of the model's original training regime. The purpose of introducing the perturbation text $x_p$ is to induce errors by tempting the model to rely on the perturbative text. This compels the model to focus more on the image content to generate correct responses, thereby reducing its dependency on linguistic cues that are not supported by visual evidence.

**Perturbation Text Design**  To ensure effectiveness and naturalness of the perturbations, we adhere to the following principles. 1) *Contextual relevance*. the perturbation is expected to be contextually relevant to the image content such that it appears to be plausible but misleading. 2) *Alignment with pretrained knowledge*. The perturbations are designed to resonate with common language priors, ensuring that they are realistic and reflect potential model biases. 3) *Semantic variation*. We ensure a diverse range of perturbations by varying the structure and thematic elements of $x_p$, aligning them with common misconceptions or biases. In practice, we use GPT-4o to generate the perturbation text. The GPT-4o model views the image, question and answer, and is instructed to construct strong and diverse perturbations based on the world knowledge as well as certain image details, without disclosing the answer. The GPT-4 instruction prompt is detailed in the appendix.

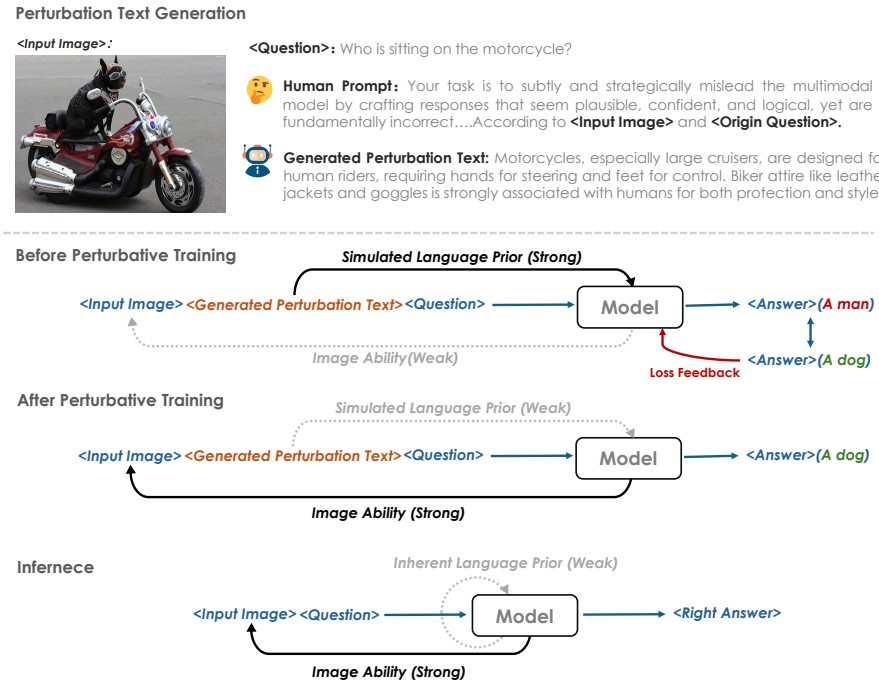

Figure 5: The generation of perturbation text and perturbative visual training of PerturboLLaVA.

| Model | No extra data generation | No extra training stage | Inference cost | Data scalability |
|---|---|---|---|---|
| Ours | ✗ | ✓ | 1× | ✓ |
| RlAIF-V | ✗ | ✗ | 1× | ✓ |
| OPERA | ✓ | ✓ | 2∼5× | ✗ |
| VCD | ✓ | ✓ | 2× | ✗ |

Table 1: Comparison of method features with other representative works.

Our approach offers significant benefits over prior approaches as summarized in Table 1. Compared to the training method like RLAIF-V, we avoid the need to collect costly preference data or train an additional reward model. Our method is easy to implement and incurs minimal additional training cost to the multimodal model, making it a nearly "free lunch" solution. The quantification of the additional training overhead is provided in the Appendix A.3. Compared to contrastive decoding strategies, our approach more fundamentally mitigates the multimodal model's excessive reliance on language priors, without introducing any additional inference overhead. Moreover, the effectiveness of our method scales with the quality and diversity of the perturbation texts.

## 4.2 MATHEMATICAL EXPLANATION

Our method can be further understood and explained from a mathematical perspective. In a more general sense, it can be interpreted as introducing additional noise or perturbation during the training phase to develop a more robust model. This approach aligns with the ideas proposed in the seminal work (Clark et al., 2019). Thus, we leverage the mathematical reasoning from that work to explain our approach. Similarly, in multimodal tasks, we define the following: $x_k$ is the $k$-th token predicted by the model, $x_{<k}$ denotes the preceding tokens in the output, $I$ refers to the corresponding image, $x_{<k}^p$ indicates the prediction based solely on language priors, while $x_{<k}^{-p}$ is the prediction made by

the multimodal model without the influence of language prior. Thus we have

$$p(x_k|x_{<k}, I) = p(x_k|x^p_{<k}, x^{-p}_{<k}, I) \tag{5}$$

$$= \frac{p(x_k|x^{-p}_{<k}, I)p(x^p_{<k}|x_k, x^{-p}_{<k}, I)}{p(x^p_{<k}|x^{-p}_{<k}, I)} \tag{6}$$

$$\propto p(x_k|x^{-p}_{<k}, I)p(x^p_{<k}|x_k, x^{-p}_{<k}, I) \tag{7}$$

$$= p(x_k|x^{-p}_{<k}, I)p(x^p_{<k}|x_k, I) \tag{8}$$

$$= p(x_k|x^{-p}_{<k}, I)\frac{p(x_k|x^p_{<k}, I)p(x^p_{<k})}{p(x_k)} \tag{9}$$

$$\propto p(x_k|x^{-p}_{<k}, I)\frac{p(x_k|x^p_{<k}, I)}{p(x_k)}. \tag{10}$$

Equations (6) and (9) are applications of Bayes' theorem. $x^p_{<k}$ and $x^{-p}_{<k}$ are mutually independent and Equation (8) follows from the conditional independence assumption.

Now take a look at the Equations (10). $p(x_k)$ models the probability of the next token. In a sufficiently uniform dataset, $p(x_k)$ can be ignored. $p(x_k|x^{-p}_{<k}, I)$ represents the behavior we expect from the multimodal model when it predicts the next token based on the image without interference from language priors. However, this probability is difficult to model directly. Instead, we designed a perturbative training that adds perturbed text before the instruction to enhance $p(x_k|x^p_{<k}, I)$, encouraging the model to learn $p(x_k|x^{-p}_{<k}, I)$. Here, $p(x_k|x^p_{<k}, I)$ represents the bias introduced by the perturbation text. Since $x^p_{<k}$ as we define it represents predictions based solely on language priors, the term $p(x_k|x^p_{<k}, I)$ can exclude the image and become $p(x_k|x^p_{<k})$, which represents the language model's prediction tendency when presented with the perturbed text. During the training of multimodal models, we can assume that the world knowledge embedded in the language model remains unchanged, so $p(x_k|x^p_{<k})$ is a perturbation term that cannot be optimized. Therefore, in the training process of multimodal models, our perturbation training method guides the model to optimize towards $p(x_k|x^{-p}_{<k}, I)$, transforming it into a fully multimodal model that is unaffected by language priors and relies entirely on image information to answer questions.

## 5 EXPERIMENTS

### 5.1 EXPERIMENTS SETTINGS

**Implementation Details**   To ensure the reliability and credibility of the results, we conducted experiments on the open-source and widely-used LLaVA1.5 dataset. By utilizing the LLaVA1.5 dataset, we fully replicated LLaVA1.5 training process and results, and use it as the baseline for our experiments. To ensure fairness, we apply perturbations directly to the LLaVA1.5 dataset rather than incorporating additional data. We conduct experiments by generating perturbed text using GPT-4o on the 160k data related to the VQA task in the dataset.

**Comparative Methods**   We selected three representative methods, each addressing hallucination from different perspectives and demonstrating strong performance: the training-based RLAIF-V (Yu et al., 2024), the decoding strategy OPERA (Huang et al., 2024) and VCD (Leng et al., 2024). For RLAIF-V, we use the open-source model weights of RLAIF-7B, which were fine-tuned on LLaVA1.5 (Liu et al., 2024b). Notably, the reward model used in this setup is LLaVA-Next 34B (Liu et al., 2024a), which may transfer some of LLaVA-Next 34B's capabilities into RLAIF-7B, potentially making the comparison with other experimental setups less fair. The hyperparameters for OPERA and VCD are provided in Section A.5 to support the reproducibility of our results. We use beam search as the default decoding strategy, with $N_{\text{beams}}$ set to 5.

We additionally evaluate six best multimodal models available today, Ovis1.6 (Lu et al., 2024b), Qwen2-VL (Wang et al., 2024), LLaVA-OneVision (Li et al., 2024), InternVL2 (OpenGVLab, 2024), Idefics3 (Laurençon et al., 2024), and MiniCPM-2.6 (Hu et al., 2024), to further verify the effectiveness of our proposed metric. These models have been trained on vast datasets, compris-

| Model | Size | Precision↑ | Recall↑ | Fscore↑ | Object↓ | Attribute↓ | Relation↓ |
|---|---|---|---|---|---|---|---|
| Ovis1.6-Gemma2 | 9B | 61.5 | 50.3 | 55.4 | 22.1 | 4.9 | 11.7 |
| Qwen2-VL | 7B | 60.8 | 50.0 | 54.9 | 24.2 | 5.1 | 9.9 |
| LLaVa-onevision | 7B | 61.3 | 48.3 | 54.1 | 22.1 | 4.9 | 11.7 |
| InternVL2 | 8B | 60.6 | 48.6 | 53.9 | 24.1 | 5.2 | 10.1 |
| Idefics3 | 8B | 59.7 | 48.2 | 53.3 | 25.1 | 6.1 | 9.1 |
| MiniCPM-2.6 | 8B | 57.3 | 47.8 | 52.1 | 26.3 | 6.7 | 9.7 |
| LLaVA1.5 | 7B | 53.3 | 45.8 | 49.2 | 28.1 | 8.1 | 10.5 |
| RLAIF-V$^\star$ | 7B | 57.7 | 47.2 | 51.9 | 25.9 | 7.0 | 9.4 |
| OPERA$^\star$ | 7B | 58.6 | 46.7 | 51.9 | 25.3 | 6.9 | 9.2 |
| VCD$^\star$ | 7B | 58.8 | 46.7 | 52.0 | 25.6 | 7.2 | 8.4 |
| Ours$^\star$ | 7B | 59.5 | 46.5 | 52.2 | 25.3 | 6.4 | 8.8 |

Table 2: Results of our method and comparison methods using our proposed hallucination measure. It should be emphasized that the models in the first part use advanced language models like Qwen2(Yang et al., 2024) and LLama3(Dubey et al., 2024), trained on extensive high-quality datasets. Later experiments marked are conducted using the LLaVA1.5 Baseline.

| Model | Size | Reward | HalFscore | | | Object HalBench | | HalBench↑ | MMB↑ | SEED↑ | CCBench↑ |
|---|---|---|---|---|---|---|---|---|---|---|---|
| | | | Prec.↑ | Recall↑ | Fscore↑ | CHAIR$_s$↓ | CHAIR$_i$↓ | | | | |
| LLaVA1.5 | 7B | ✗ | 53.3 | 45.8 | 49.2 | 54.2 | 15.0 | 46.9 | 67.3 | 65.3 | 29.4 |
| VCD | 7B | ✗ | 58.8 | 46.7 | 52.0 | 51.2 | 14.3 | 46.9 | 66.2 | 64.7 | 29.4 |
| OPERA | 7B | ✗ | 58.6 | 46.7 | 51.9 | 50.8 | 14.2 | 47.1 | 67.4 | 65.3 | 29.4 |
| RLAI-F | 7B | LLaVA-Next | 57.7 | **47.2** | 51.9 | **18.1** | **4.7** | **51.3** | 63.7 | 65.4 | 27.8 |
| Ours | 7B | ✗ | 59.5 | 46.5 | 52.2 | 36.1 | 10.4 | 47.5 | 68.9 | 65.6 | 30.6 |
| OPERA+Ours | 7B | ✗ | **60.2** | 47.0 | **52.8** | 33.1 | 10.1 | 47.6 | **68.9** | **65.6** | **31.0** |

Table 3: Results of our method and comparative methods on different benchmarks. The best result is marked in bold, and the second-best result is underlined, respectively.

ing hundreds of millions of samples, and required significant computational resources, including extensive GPU hours. Their general performance substantially surpasses that of LLaVA1.5.

## 5.2 MAIN RESULTS

**HalFscore Evaluation** Table 2 shows the our final HalFscore includes precision, recall, and fscore, and further classifies hallucinations into object, attribute, and relation types for more detailed evaluation. From this metric, we observe that compared to the LLaVA1.5 baseline, the leading methods have achieved substantial and comprehensive improvements in both precision and recall. Notably, there is a significant reduction in hallucinations when describing objects and attributes, although there remains room for improvement in relation hallucinations. Our method also demonstrates marked improvement over LLaVA1.5, with a +6.2 increase in precision, +0.7 in recall, and +2.4 in Fscore. In comparisons with OPERA, VCD, and RLAIF-V, our approach achieved the best results in precision and fscore, though the improvement in recall is less pronounced. Overall, by leveraging these hallucination mitigation techniques, OPERA, VCD, and RLAIF-V have achieved recall scores comparable to MiniCPM-2.6, while our method has even surpassed MiniCPM, further underscoring the effectiveness of our approach.

**CHAIR Evaluation** To evaluate caption-based object hallucination, we use the CHAIR metric (Rohrbach et al., 2018), which measures the percentage of generated words that correspond to actual objects in the image. The results in Table 3 show that, on this metric, our method significantly reduces hallucinations compared to the LLaVA1.5 baseline and outperforms both OPERA and VCD by a large margin. However, RLAIF-V performs better on this metric, showing fewer object hallucinations. A possible reason is that it uses additional feedback model. Moving forward, we plan to design targeted perturbation texts for objects to further enhance performance.

| Type | Example |
|---|---|
| **Version1** | `<image>` What follows presents incorrect guidance based on the image. `<perturbation text>` Please ignore any false guidance and focus on the image to answer the question below. `<Question>` |
| **Version2** | `<image>` `<perturbation text>` Please examine the image and respond to the question below. `<Question>` |
| **Version3** | `<image>` `<perturbation text>` `<Question>` |
| **Random** | `<image>` Coming up is a portion of non-visual text. `<random perturbation text>` Ignore the previous perturbation text and focus on the image to complete the task. `<Question>` |

Table 4: Different variants of perturbative visual training.

| Type | HalFscore↑ | | | Object HalBench↓ | | | HalBench↑ | MMB↑ | SEED↑ | CCBench↑ |
|---|---|---|---|---|---|---|---|---|---|---|
| | Precision | Recall | Fscore | CHAIR$_s$ | CHAIR$_i$ | length | | | | |
| LLaVA1.5 | 53.3 | 45.8 | 49.2 | 54.2 | 15.0 | 100 | 46.9 | 67.3 | 65.3 | 29.4 |
| Version1 | 59.5 | 46.5 | 52.2 | 36.1 | 10.4 | 89 | 47.5 | 68.9 | 65.6 | 30.6 |
| Version2 | 59.6 | 46.0 | 52.0 | 33.6 | 10.2 | 82 | 49.2 | 67.8 | 65 | 29.2 |
| Version3 | 59.7 | 46.1 | 52.0 | 32.3 | 10.6 | 78 | 49.6 | 66.5 | 64.7 | 28.8 |
| Random | 57.1 | 45.6 | 50.7 | 52.4 | 15.0 | 99 | 49.0 | 67.4 | 65.1 | 29.4 |

Table 5: Effect of varying perturbation levels during training.The best results are underlined.

**HallusionBench Evaluation** HallusionBench (Guan et al., 2024) evaluates how well multi-modal models handle language hallucinations and visual illusions. Previous hallucination assessment mainly focused on dense caption tasks, and by incorporating HallusionBench, we aim to enhance our hallucination evaluation through the addition of multiple-choice and vision reasoning tasks. The results in Table 3 demonstrate a substantial improvement (+0.6) of our method on the Hallusion-Bench metric, whereas the VCD decoding strategy merely maintained the baseline, OPERA led to a modest increase of 0.2. This outcome further confirms that our method is not only effective in dense captioning tasks but also excels in vision reasoning tasks.

**General Multimodal Ability** We assess the models' general ability on three widely used benchmarks: MMBench (Liu et al., 2023c), CCBench (Liu et al., 2023c), and SEEDImage (Li et al., 2023a), to verify that our model's general capabilities. From the Table 3, we observe that both VCD and RLAIF-V result in some degradation in general performance, while OPERA remains stable. In contrast, our method not only avoids degradation in general metrics but also shows improvements across multiple areas, with gains of +1.6 in MMB, +0.3 in SEEDImage, and +1.2 in CCBench. We attribute this improvement in generalization to perturbation training, which compels the multimodal model to more effectively leverage image information.

## 5.3 ANALYSES

**Impact of the Perturbation Strengths** To assess the effect of perturbation levels on training, we devised three methods of inserting perturbation text, detailed in Table 4. The first method alerts the model to upcoming text perturbations, instructing it to ignore them. The second focuses the model on image regions without referencing perturbations. The third offers no guidance. These methods progressively increase the level of perturbation and the difficulty of training the multimodal model. As Table 5 shows, increasing perturbation levels enhances training difficulty and reduces hallucinations, as indicated by HalFscore and CHAIR metrics. However, this also leads to shorter captions and lower recall, suggesting a more cautious model behavior. General performance metrics decline consistently, implying that while perturbations decrease hallucinations, excessive perturbation use may impair overall performance.

**Impact of the Perturbation Relevance** To investigate the impact of relevance of perturbation text, we conducted experiments using randomly sampled texts from a text dataset instead of the

| Auto Evaluation | Human Evaluation | Pearson Correlation |
|---|---|---|
| Recall | Recall_h | 78.1 |
| Precision | Precision_h | 80.7 |
| MMhalBench | Precision_h | 71.7 |

Table 6: Correlations between automatically computed metrics and human subjective opinions.

carefully designed perturbations used earlier. The results in Table 5 show that introducing random text perturbations, compared to the LLaVA1.5 baseline, still mitigates hallucinations in multimodal models, as evidenced by improvements in HaFscore and HalBench. However, the effectiveness of random perturbations is inferior to that of targeted text-based experiments. We suggest that even random perturbations influence the training of multimodal models, as the training is disrupted by the random text interference. However, more relevant perturbations exert a stronger disruptive effect, prompting the model to focus more on the image to overcome these challenges.

**Human Analysis of HalFscore**   We perform the user study to explore whether the proposed HalF-score correlates well with human evaluation. We show an image and ask human raters to compare two paragraphs generated and determine which generated text describes the main subject and detailed information of the input image more comprehensively (recall_h), and which text contains less information that is not present in the input image (precision_h). Four methods, LLaVA 1.5, MiniCPM-2.6, Internvl2, and our method are selected for human evaluation. Each method is compared with other methods for 12 times. We compute the human evaluated scores for the recall and precision and drive the Pearson correlations coefficients between the scores of human evaluation and that of HalFscore. The results are shown in the Table 6. We observe that our precision score and recall score align with human evaluations. Specifically, we compare our method with another metric, MMhalbench. The correlation coefficient between the metric and the human evaluations of precision scores is relatively smaller than HalFscore. The comparison demonstrates our metric aligns with human evaluations more than directly asking large language models to rate the level of hallucination. Further details about the user study can be found in Section A.7 of the Appendix.

**Complementary Effect to Decoding Strategies.**   PerturboLLaVA is a novel training strategy that enhances model performance in the SFT phase. It can be integrated seamlessly with post-SFT optimization RLAIF-V and OPERA during inference. In our tests, using OPERA as a plugin with PerturboLLaVA during inference led to performance boosts, as detailed in Table 3. PerturboLLaVA with OPERA achieved further gains, improving HalFscore by 0.6, with a 3-point improvement in the $CHAIR_s$. Hallucination reasoning in Hallusionbench also improved by +0.1, while general capability increased by 0.4 in CCBench. Compared to applying OPERA on LLaVA1.5, replacing the baseline model with PerturboLLaVA resulted in improved performance in both hallucination reduction and general capability. Thus, PerturboLLaVA introduces a new optimization direction that complements existing strategies, offering an additional benefit.

## 6   CONCLUSION

In our work, we introduce a concept-level HalFscore that enables fine-grained analysis of various hallucinations involving objects, attributes, and relations in dense captions. This metric also reflects the overall captioning capability of multi-modal models. To mitigate hallucinations, we propose a simple yet effective training strategy that guides multi-modal models to reduce reliance on linguistic priors and prioritize image information. Our method outperforms leading state-of-the-art approaches without incurring additional training or inference overhead. Overall, we believe that our proposed metric enhances the evaluation of hallucinations in multi-modal models and that our method effectively mitigates hallucination issues. The proposed method shows great promise of becoming a standard strategy for training robust multimodal models.

## ACKNOWLEDGMENTS

This work was supported by the National Key R&D Program of China (No. 2022ZD0160101), the Ningbo Science and Technology Bureau (Grant Number 2024Z291), and the National Natural

Science Foundation of China (No. 62206244). CC and ML contributed equally. Part of the work was done when CC was doing an internship at WeChat Group.

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

# A   APPENDIX

## A.1   IMPLEMENTATION DETAILS OF HALFSCORE

### A.1.1   STATISTIC OF IMAGE-CAPTION PAIR

| Statistics | Number | Ratio |
|---|---|---|
| Indoor Scene | 307 | 30.7% |
| Public Spaces | 134 | 13.4% |
| Home | 79 | 7.9% |
| Stores | 49 | 4.9% |
| Office | 27 | 2.7% |
| Others | 18 | 1.8% |
| Outdoor Scene | 693 | 69.3% |
| Urban | 194 | 19.4% |
| Architecture | 145 | 14.5% |
| Transportation | 152 | 15.2% |
| Natural Scenery | 128 | 12.8% |
| Rural | 68 | 6.8% |
| Others | 6 | 0.6% |
| Total Number of Image | 1,000 | 100% |

Table 7: Statistic of our image data.

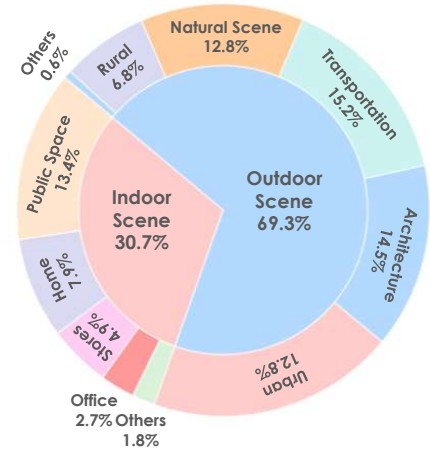

Table 8: Distribution of our image data.

We manually selected 1,000 images from the DCI dataset to ensure that the final image data used is characterized by high quality and diversity. The images we ultimately used consist of approximately 30.7% indoor scenes and 69.3% outdoor scenes, and we further subdivided both indoor and outdoor scenes into more detailed categories. The statistical results are shown in the figure and table. By using diverse image data, we ensure comprehensive evaluation of the multimodal model's hallucinatory outputs across various scenarios.

### A.1.2   CONSTRUCTION OF CONCEPT GRAPH

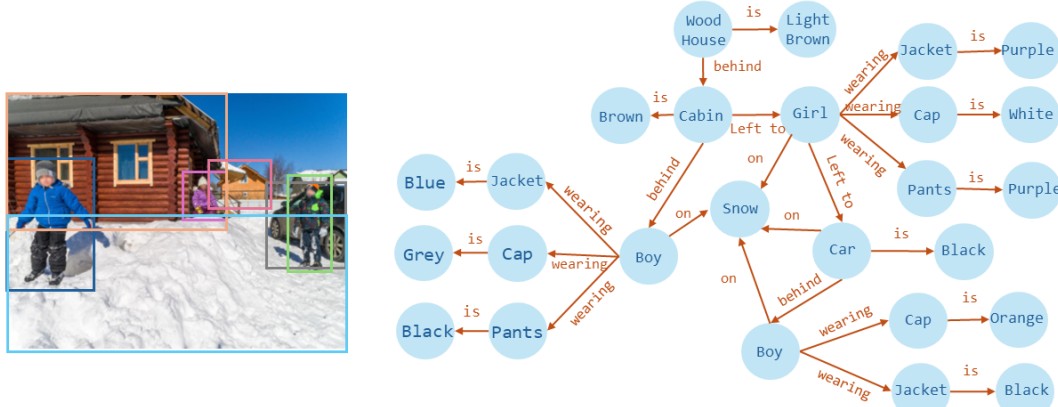

Figure 6: Example of the construction of concept graph.

We demonstrate how a caption is parsed into a concept graph within our evaluation pipeline. First, we utilize triplets to extract several concept groups from the original caption[7], which are organized in the form of subjects, objects, and the relationships between them. These triplets efficiently capture information from the caption and are sufficiently general to represent object relationships[8], such as `<Boy, behind, Cabin>`,`<Girl, left to, Car>`, as well as attributes like `<jacket, is, blue>`, and states or actions such as `<Girl, Wearing, Pants>`. This demonstrates the robustness of triplet-based information extraction. Once the caption is transformed into triplets,

**VLM Caption**

Three children are playing in the snow in front of a log cabin. One child is wearing a blue jacket, another is wearing a pink jacket, and the third child is wearing an orange jacket. They are all wearing hats, with one child wearing an orange hat and the other two wearing blue hats.There is a car parked in front of the cabin, and the children seem to be enjoying themselves as they play in the snow.

Figure 7: VLM Caption

**VLM Triplets**

```
1.  ("object"||CHILDREN||are||three)
2.  ("object"||CHILDREN||on||Snow)
3.  ("object"||LOG CABIN||behind||boy)
4.  ("object"||CHILD||wearing||blue jacket)
5.  ("object"||CHILD||wearing||pink jacket)
6.  ("object"||CHILD||wearing||orange jacket)
7.  ("object"||HATS||all children||wearing)
8.  ("object"||HAT||is||orange)
9.  ("object"||HATS||is||blue)
10. ("object"||CAR||parked in front of||cabin)
11. ("object"||CHILDREN||enjoying||themselves)
12. ("relationship"||CHILDREN||SNOW||The children are
playing in the snow||9)
14. ("relationship"||CHILD||BLUE JACKET||One child is
wearing a blue jacket||8)
15. ("relationship"||CHILD||PINK JACKET||One child is
wearing a pink jacket||8)
16. ("relationship"||CHILD||ORANGE JACKET||One child is
wearing an orange jacket||8)
17. ("relationship"||CHILDREN||HATS||All children are
wearing hats||8)
18. ("relationship"||CHILD||ORANGE HAT||One child is wearing
an orange hat||8)
19. ("relationship"||CHILDREN||BLUE HATS||Two children are
wearing blue hats||8)
20. ("relationship"||CAR||LOG CABIN||The car is parked in
front of the log cabin||7)
21. ("relationship"||CHILDREN||ENJOYING THEMSELVES||The
children seem to be enjoying themselves as they play in the
snow||9)
```

Figure 8: VLM Triplets

we further match these triplets based on the subject and object9. Identical subjects or objects are represented as nodes in the concept graph, while the relations from the original triplets are transformed into edges within the graph. The final concept graph, as shown in the figure, consists of entity concepts as nodes and relation concepts as directed edges. This graph effectively organizes both concrete and abstract concepts from the caption in a structured format, enabling subsequent hallucination and omission analysis.

### A.1.3 ANALYSIS OF HALLUCINATION

Here we provide an example of our generated analysis of halluciation10 and omission11.

```
VLM Graph
1.("NODE":CHILDREN||"EDGE":are||"NODE":three)
2.("NODE":CHILDREN||"EDGE":on||"NODE":SNOW)
3.("NODE":LOG CABIN||"EDGE":behind||"NODE":BOY)
4.("NODE":CHILDREN||"NODE":SNOW||"EDGE":The children are
playing in the snow||9)
5.("NODE":CHILD||"NODE":BLUE JACKET||"EDGE":One child is
wearing a blue jacket||8)
6.("NODE":CHILD||"NODE":PINK JACKET||"EDGE":One child is
wearing a pink jacket||8)
7.("NODE":CHILD||"NODE":ORANGE JACKET||"EDGE":One child is
wearing an orange jacket||8)
8.("NODE":CHILDREN||"NODE":HATS||"EDGE":All children are
wearing hats||8)
9.("NODE":CHILD||"NODE":ORANGE HAT||"EDGE":One child is
wearing an orange hat||8)
10.("NODE":CHILDREN||"NODE":BLUE HATS||"EDGE":Two children
are wearing blue hats||8)
11.("NODE":CAR||"NODE":LOG CABIN||"EDGE":The car is parked in
front of the log cabin||7)
12.("NODE":CHILDREN||"NODE":ENJOYING THEMSELVES||"EDGE":The
children seem to be enjoying themselves as they play in the
snow||9)
```

Figure 9: VLM Graph

## A.2 IMPLEMENTATION DETAILS OF PERTURBOLLAVA

**Construction of perturbation text**  Using GPT-4o, we generated the perturbation text. The prompt guided GPT-4o to act as an expert multimodal attacker, producing text based on image elements and aligned with world knowledge, aimed at causing the multimodal model to deviate from the ground truth answer. The perturbation text was constructed through two rounds of Q&A, with the first-round prompt, as shown in the figure, designed to guide GPT-4o in generating the perturbation text as required. Specifically, the construction of hallucination text was divided into five steps: the first step involves analyzing the image, question, and correct answer, focusing on key regions of the image; the second and third steps ensure that GPT-4o adheres to the guidelines outlined in our methodology; the fourth step refines the previous text; and the final step checks the text length, ensuring that the perturbation text is sufficiently detailed and complex before outputting the final version. The second round of Q&A serves as a validation and supplement to the text output from the first round. The specific text for the second round is shown in the figure.

**Insertion of perturbation text**  The SFT format of LLaVA1.5 is structured as multi-turn conversations. We insert the generated perturbation text before the first-round question to ensure that the multimodal model is exposed to the perturbation information during each Q&A turn. For the insertion, we include prompts to signal to the multimodal model that this is a hallucination. The specific format is: `<image><hint prompt1><perturbation text><hint prompt2>`.

`<hint prompt1>` informs the model that the following text is a hallucination perturbation, while `<hint prompt2>` prompts the model to disregard the previous perturbation and focus on answering the question based on the image. To prevent the multimodal model from overfitting to a fixed hint prompt pattern, we did not use a static `<hint prompt>`. Instead, we generated multiple variations of `<hint prompt1>` and `<hint prompt2>`, randomly selecting one for each instance to avoid overfitting.

**Examples of perturbation text**  Here are two perturbed texts sampled from our training data, specifically constructed based on GPT-4o's view of the image, question, and answer. These texts are based on certain image elements and align with world knowledge but contradict the actual image

**Halluciation Analysis**

**Analysis:**
- **Entry 5 in VLM List:**
- The GT List does not mention any child wearing a pink jacket. This counts as one halluci­nation for the incorrect object "CHILD" with the attribute "wearing pink jacket".
- **Entry 6 in VLM List:**
- The GT List does not mention any child wearing an orange jacket. This counts as one hallucination for the incorrect object "CHILD" with the attribute "wearing orange jacket".
- **Entry 8 in VLM List:**
- The GT List does not mention any child wearing an orange hat. This counts as one hallu­cination for the incorrect object "HAT" with the attribute "orange".
- **Entry 9 in VLM List:**
- The GT List does not mention any child wearing blue hats. This counts as one hallucination for the incorrect object "HATS" with the attribute "blue".
- **Entry 10 in VLM List:**
- The GT List mentions a "DARK-COLORED CAR" along the right side and behind the boy on the right, not in front of the cabin. This counts as one hallucination for the incorrect object "CAR" with the attribute "parked in front of cabin".
- **Entry 11 in VLM List:**
- The GT List does not mention the children "enjoying themselves". This counts as one hallucination for the incorrect attribute "enjoying themselves".
- **Entry 4 in VLM List:**
- "CHILD" with "PINK JACKET" is an incorrect object already counted. This relationship involves an incorrect object, so it does not count separately.
- **Entry 5 in VLM List:**
- "CHILD" with "ORANGE JACKET" is an incorrect object already counted. This relation­ship involves an incorrect object, so it does not count separately.
- **Entry 7 in VLM List:**
- "HAT" with "ORANGE" is an incorrect object already counted. This relationship involves an incorrect object, so it does not count separately.
- **Entry 8 in VLM List:**
- "HATS" with "BLUE" is an incorrect object already counted. This relationship involves an incorrect object, so it does not count separately.
- **Entry 9 in VLM List:**
- "CAR" with "parked in front of cabin" is an incorrect object already counted. This rela­tionship involves an incorrect object, so it does not count separately.
- **Entry 10 in VLM List:**
- "CHILDREN" with "enjoying themselves" is an incorrect object already counted. This relationship involves an incorrect object, so it does not count separately.nnIncorrect Serial Numbers: 5, 6, 8, 9, 10, 11

Figure 10: Analysis of Halluciation

content, aiming to induce the multimodal model to fall into the trap of relying on language priors during training.

A.3    QUANTIFYING THE ADDITIONAL TRAINING OVERHEAD OF PERTURBOLLAVA

| Cost of Training | Average Memory Cost (GB) | Training Time cost (min) |
|---|---|---|
| **Baseline** | 62.3 | 264 |
| **PerturboLLaVA** | 63.8 | 281 |
| **Additional Overhead Ratio** | 2.6% | 6.4% |

Table 9: Comparison of Training Costs for Baseline and PerturboLLaVA

We quantified the additional GPU memory and training time overhead introduced by incorporating perturbation text into the training data. Compared to the original SFT stage, our method results in

---

**Omission Analysis**

**Analysis**:
- **Entry 2 in GT List:**
- The object "LEFT SIDE OF LOG CABIN" is missing from the VLM List. This counts as one missing object.
- **Entry 4 in GT List:**
- The object "WINDOW FRAMES" is missing from the VLM List. This counts as one missing object.
- **Entry 5 in GT List:**
- The object "ROOFTOP OF LOG CABIN" is missing from the VLM List. This counts as one missing object.
- **Entry 10 in GT List:**
- The object "CHIMNEY" is missing from the VLM List. This counts as one missing object.
- **Entry 11 in GT List:**
- The object "YARD OF LOG CABIN" is missing from the VLM List. This counts as one missing object.
- **Entry 15 in GT List:**
- The object "BLACK PANTS" is missing from the VLM List. This counts as one missing object.
- **Entry 20 in GT List:**
- The object "PILE OF SNOW" is missing from the VLM List. This counts as one missing object.
- **Entry 21 in GT List:**
- The object "BOY ON THE RIGHT" is missing from the VLM List. This counts as one missing object.
- **Entry 22 in GT List:**
- The object "BLACK AND ORANGE BEANIE CAP" is missing from the VLM List. This counts as one missing object.
- **Entry 23 in GT List:**
- The object "BLUE PANTS" is missing from the VLM List. This counts as one missing object.
Missing Serial Numbers: 2, 4, 5, 10, 11, 15, 20, 21, 22, 23

Figure 11: Analysis of Omission

---

only a 2.6% increase in memory consumption and a 6.5% increase in training time, highlighting the efficiency of our approach.

## A.4 ADDITIONAL QUALITATIVE RESULTS

## A.5 EXPERIMENTS DETAILS

| | Pretrain | SFT |
|---|---|---|
| Traing data | LLaVA-Pretrain 558k | LLaVA-SFT 665k |
| Trainable module | Projector | Projector & LLM Backbone |
| Learning rate | $1.0 \times 10^{-3}$ | $2.0 \times 10^{-5}$ |
| LR scheduler | CosineAnnealing | CosineAnnealing |
| Warmup ratio | 0.03 | 0.03 |
| Training epochs | 1 | 1 |
| Global Batch size | 256 | 128 |
| Sequence length | 2048 | 2048 |
| Optimizer | AdamW | AdamW |

Table 10: Training settings for LLaVA1.5 reproduction of pretrain and finetune.

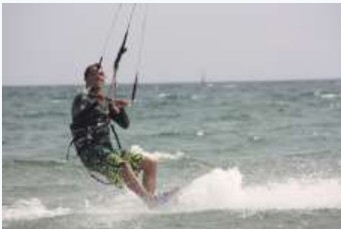 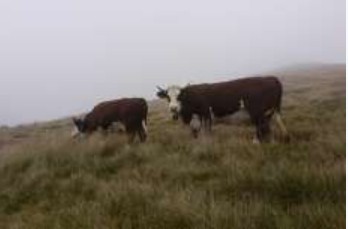

**Question 1:**
Is the person in the image surfing or kiteboarding?
**Answer 1:**
The person in the image is kiteboarding, which involves riding a surfboard while holding onto a handle connected to a rope and a parachute-like kite.

**Question 2:**
What is the person doing to glide on the water?
**Answer 2:**
The person is holding onto a handle connected to a rope, which is attached to a kite. The wind catches the kite, providing the necessary force to propel the kiteboarder across the water.

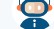 **Perturbation Text:**

Surfing is a water sport in which individuals ride on the face of a moving wave heading toward the shore, typically using a surfboard. Essential elements of surfing include a suitable wave, a surfboard, and a surfer who possesses the skills to paddle out to the wave, stand on the board as the wave lifts it, and ride the wave while maintaining balance. The surfer must also be able to read the wave, anticipate its movement, and navigate any obstacles such as rocks or other surfers. Safety gear like leashes and an awareness of the marine environment, including currents and potential marine life hazards, are also crucial. (Perturbation Text 1) In water sports, to glide effectively, an individual requires a buoyant device such as a surfboard or kayak, must don a personal flotation device for safety, and acquire the fundamental skills for maintaining balance and navigating the water. Additionally, it is imperative to have knowledge of the local water conditions and to adhere to safety protocols. (Perturbation Text 2)

**Question 1:**
What color are the cows in the image?
**Answer 1:**
The cows in the image are brown and white.

**Question 2:**
What are the cows doing in the image?
**Answer 2:**
The cows are grazing and relaxing in the grassy field.

**Perturbation Text:**

Cows, commonly associated with dairy cattle of the species Bos taurus, are often characterized by their black and white patches, as seen in Holsteins. However, their coats can display a range of colors, including solid white, red, and other hues, with variations depending on the breed. For example, the Jersey breed tends to have a fawn coloration, and Angus cattle are typically black.(Perturbation Text 1) Cows within a dairy farming context are typically involved in activities that are essential to agricultural operations. Their primary activity for maintaining their health and producing milk. During milking times, cows are guided into a milking parlor where they stand quietly to be milked, either by a farmer or an automated system. Resting is also a part of their daily routine, as cows need downtime to regenerate and maintain their energy. Occasionally, cows may be loaded onto vehicles for transportation to different locations such as markets, breeding facilities, or veterinary clinics. (Perturbation Text 2)

Figure 12: Examples of additional perturbation text.

**Reproduction of LLaVA1.5**  Using the official LLaVA1.5 open-source data and the XTuner framework, we successfully reproduced LLaVA1.5. The training data, trainable modules, and optimizer parameters are detailed in the Table 10, to ensure the reproducibility of our experiments.

**Evaluation of Object HalBench**  We set the random seed to 0 and randomly select 1,000 images from the MSCOCO2014 validation set as the ground truth for Object HalBench. The query for all models is "Please describe this image in detail." We use beam search as the decoding strategy for all models, with $N_{beams}$ set to 5, to ensure the experiments are deterministic and consistent across models.

**Evaluation of HalFscore**  To ensure that the descriptive behavior is the model's spontaneous output, we continue to use "Please describe this image in detail." in the HalFscore evaluation to perform

**VCD**

In the image, a police officer is riding a white motorcycle with a red flag attached to the back. The officer is wearing a green vest and a helmet for safety. The motorcycle is positioned in the middle of the scene, and the officer appears to be riding down a street. There are a few other people visible in the background, but they are not the main focus of the scene. Additionally, a cell phone can be seen in the top right corner of the image, possibly belonging to one of the people present.

**OPERA**

The image features a police officer riding a white motorcycle with a red flag attached to the back. The officer is wearing a helmet for safety while riding down the road. The motorcycle is positioned in the center of the scene, and the officer appears to be actively patrolling or responding to a situation. In the background, there is a car parked on the side of the road, and a person can be seen standing nearby, possibly observing the police officer or going about their day. The scene conveys a sense of law enforcement presence and vigilance in maintaining road security.

**RLAIF-V**

The image features a person in a safety jacket riding a motorcycle on a road. The motorcycle has a red flag attached to the front, making it stand out. The rider is likely a police officer or someone who is visible and noticeable to others. The motorcycle takes up a significant portion of the image, showing its features and the person riding it. The scene gives off an impression of a person on a mission or patrolling, due to the safety jacket and the prominent motorcycle. The focus on the rider and the motorcycle highlights the main subject in the image.

**PerturboLLaVA (Ours)**

A man is riding a motorcycle down a road. The motorcycle has a red flag attached to the front, and the man is wearing a green vest and a helmet. The man appears to be a police officer, as he is riding a police motorcycle.

Figure 13: [Additional Qualitative Result 1] Among the four models, *PerturboLLaVA (Ours)* offers a concise and clear description, focusing on key elements like the police officer, green vest, and red flag on the motorcycle. Unlike VCD and OPERA, which mention unnecessary details such as a cell phone, *PerturboLLaVA* highlights the main subject without distraction. While RLAIF-V speculates about the officer's identity, *PerturboLLaVA* confidently identifies the scene, demonstrating superior accuracy and clarity.

**VCD**

The image captures a soccer game in progress, with a man in a red jacket kicking a soccer ball towards a goalie. The goalie is positioned in front of a soccer net, ready to catch the ball. There are several other people in the scene, likely teammates and opponents, scattered around the field. Some of them are closer to the goalie, while others are further away. In total, there are eight people visible in the image, including the man kicking the ball and the goalie.

**OPERA**

The image depicts a group of soccer players on a field, engaged in a game. One of the soccer players is preparing to kick a soccer ball, which is located near the center of the field. There are two other soccer balls scattered around the field, indicating that the game is in progress. In addition to the soccer players, there are a few chairs placed around the field, possibly for spectators or resting players. The scene captures the excitement and action of a soccer game, with players actively participating and enjoying the sport.

**RLAIF-V**

The image captures a dynamic soccer moment with several players on the field. The main focus is on a player in a red jacket, who is kicking a soccer ball towards the net. Several other players are present in the scene, likely teammates and opponents, spread across the field. Numerous soccer balls can be spotted on the lush green field, indicating an active and ongoing match. The net is visible in the foreground, adding to the sense of a well-defined playing area. The composition of the image conveys the action-packed and exciting nature of the soccer game, showcasing the players' skills and the fast-paced nature of the sport.

**PerturboLLaVA (Ours)**

A soccer goalie wearing a green jersey stands in front of the goal, ready to make a save. The goalie is surrounded by three soccer balls, which are red and white. The balls are scattered around the goal area, with some closer to the goalie and others further away. In addition to the goalie, there are a few other people in the scene, likely teammates or opponents. They can be seen on the left side of the image.

Figure 14: [Additional Qualitative Result 2] *PerturboLLaVA (Ours)* excels with its accurate depiction, correctly identifying key elements like the goalie and the exact number of soccer balls (three), while avoiding false details in other descriptions. VCD and RLAIF-V incorrectly mention "eight people" or "numerous soccer balls", and OPERA similarly errs with "two other soccer balls" and irrelevant chairs.

the dense captioning task. Similar to Object HalBench, we use beam search as the decoding strategy for all models, with $N_{beams}$ set to 5.

**VCD**
In the image, a man and a woman are walking down a sidewalk near a train station. The man is carrying a large backpack, while the woman is pulling a suitcase behind him. They appear to be travelers, possibly arriving at or departing from the train station. There are several other people in the scene, some closer to the train station and others further away. A clock can be seen in the background, indicating the time for the travelers. A car is also visible in the scene, parked near the train station.

**OPERA**
The image depicts a man and a woman walking down a sidewalk near a train station. The man is carrying a backpack and a suitcase, while the woman is also carrying a suitcase. They appear to be travelers, possibly arriving at or departing from the train station. There are several other people in the scene, some closer and others further away from the train station. A clock can be seen in the background, indicating the time for the travelers. Additionally, there is a car parked near the train station, possibly belonging to one of the travelers.

**RLAIF-V**
In the image, a train can be seen in the background, indicating a possible train station. A man is prominently walking down a walkway or platform, carrying several pieces of luggage, including black bags. A clock is also visible in the background, likely indicating the time for passengers. There are several other people present in the area, possibly waiting for their trains or moving about the station. The scene gives off a sense of a bustling and active train station where people are going about their travels with their belongings.

**PerturboLLaVA (Ours)**
A man and a woman are walking down a sidewalk next to a train. The man is carrying a blue backpack and a black bag. The woman is also carrying a backpack. There are several other people in the scene, some of whom are also carrying backpacks. In the background, there is a clock visible. A woman can be seen far away, probably just depart from the train.

**Input Image**

**Instruction**
Please describe this image in detail.

Figure 15: [Additional qualitative results 3] *PerturboLLaVA (Ours)* surpasses the other models by accurately depicting key details, such as "the man carrying a blue backpack and a black bag, while the woman carries a backpack", avoiding the incorrect mention of a suitcase by VCD, OPERA, and RLAIF-V. Additionally, *PerturboLLaVA* avoids false details like "the car near the train station" mentioned by VCD and OPERA. It also correctly identifies the presence of other people carrying backpacks, providing a more precise and coherent scene description

**Hyperparameters of OPERA**  We directly used the open-source implementation of OPERA, with the OPERA strategy based on beam search. The beam search parameters are the same as those for our model, with $N_{beams}$ set to 5. Other OPERA parameters are set as Table 11 shows. This configuration leads to higher time consumption but also yields better performance.

| Parameter | Value |
|---|---|
| $N_{beams}$ | 5 |
| Scale factor | 50 |
| Threshold | 15 |
| Number of attention candidates | 5 |
| Penalty weights | 1 |

Table 11: OPERA parameter settings used in our experiments

**Hyperparameters of VCD**  Given that the open-source implementation of VCD is tailored for sample search, while our default decoding strategy is beam search, we developed a beam search version of VCD based on the original VCD code. The specific parameters for VCD are provided in the Table 12. To ensure the reproducibility of our results, we will also release our VCD implementation.

| Parameter | Value |
|---|---|
| $N_{beams}$ | 5 |
| Image noise steps $T$ | 999 |
| VCD $\alpha$ | 0.5 |
| VCD $\beta$ | 0.1 |

Table 12: VCD parameter settings used in our experiments

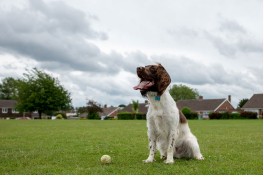

**VCD**
In the image, a brown and white dog is sitting on a lush green field, *holding a tennis ball in its mouth*. The dog appears to be a spaniel, and it seems to be enjoying its time outdoors. *There are several cars parked in the background*, indicating that the field might be close to a parking area or a road. *Additionally, there are two other people visible in the scene, possibly observing the dog or engaging in their own activities.*

**OPERA**
The image features a brown and white dog standing on a lush green field, *holding a tennis ball in its mouth*. The dog appears to be a spaniel, and it seems to be enjoying its time outdoors. In the background, *there are several cars parked along the edge of the field*, indicating that this might be a popular spot for people to gather and play with their pets. *The cars are spread out, with some closer to the foreground and others further in the background, creating a sense of depth in the scene.*

**RLAIF-V**
The image features a cute English Springer Spaniel dog standing in a lush, green field on a cloudy day. The dog is prominently positioned in the field, *and a ball can be spotted nearby*, indicating that it was likely playing. The cloudy sky sets a pleasant backdrop for the scene. *In the background, there are houses visible, suggesting that the field may be located near a residential area.* The overall atmosphere evokes a sense of outdoor fun and leisure with the adorable dog as the main subject.

**PerturboLLaVA (Ours)**
The image features a brown and white dog standing in a grassy field. The dog appears to be enjoying the outdoors, with its mouth open. *There is a ball nearby. There are several cars parked in the background, likely belonging to the people who live in the nearby houses.* The dog is positioned in the middle of the field, making it the focal point of the scene.

**Input Image**

**Instruction**
Please describe this image in detail.

Figure 16: [Additional qualitative result 4] *PerturboLLaVA (Ours)* excels over the other models by offering a clear and accurate description of the scene, focusing on the key elements such as "the brown and white dog standing in the field". Unlike VCD and OPERA, which incorrectly state that "the dog is holding a tennis ball in its mouth", *PerturboLLaVA* avoids this error. Additionally, it correctly identifies the cars in the background, avoiding VCD's mistaken mention of two other people and OPERA's incorrect description of "cars spread out to create depth".

## A.6 UNDERSTANDING THE MECHANISM OF PERTURBOLLAVA FROM ANOTHER PERSPECTIVE

We use $\mu(x) = \prod \mu(x_t \mid x_{<t})$ to represent a pure language model, and $\pi_\theta(x \mid x_{<t}, I)$ to represent our multimodal model. We initialize the multimodal model with $\mu(x \mid x_{<t})$. Additionally, we treat $\mu(x)$ as the language prior of the multimodal model.

When comparing the gradient difference with the original LLM, we obtain a ratio for DPO:

$$r_\theta(x, I) = \log \frac{\pi_\theta(x \mid I)}{\mu(x)} = \sum_{t=1} \log \frac{\pi_\theta(x_t \mid x_{<t}, I)}{\mu(x_t \mid x_{<t})}.$$

After introducing the perturbed text $x^+$, we induce $\mu(x)$ to leverage the language prior to $\mu(x, x^+)$, which increases the gradient for model training:

$$\log \frac{\pi_\theta(x_t \mid x_{<t}, x^+, I)}{\pi_\theta(x_t \mid x_{<t}, I)} = \log \frac{\pi_\theta(x_t \mid x_{<t}, x^+, I)}{\mu(x_t \mid x_{<t}, x^+)} \cdot \frac{\mu(x_t \mid x_{<t})}{\pi_\theta(x_t \mid x_{<t}, I)} \cdot \frac{\mu(x_t \mid x_{<t}, x^+)}{\mu(x_t \mid x_{<t})}.$$

This simplifies to:

$$r_\theta(x_t \mid x_{<t}, x^+, I) - r_\theta(x_t \mid x_{<t}, I) + \log \frac{\mu(x_t \mid x_{<t}, x^+)}{\mu(x_t \mid x_{<t})}.$$

This also implies that the optimal perturbed text should maximize the gap of $\frac{\mu(x_t \mid x_{<t}, x^+)}{\mu(x_t \mid x_{<t})}$.

## A.7 DETAILS OF USER STUDY

In our implementation, four methods, LLaVA 1.5, MiniCPM-2.6, Internvl2, and our method are selected for user study. Each method is compared with other methods for 12 times. We use the Bradley-Terry model (Bradley & Terry, 1952), a statistical model for pairwise comparisons between items or individuals, to obtain the abilities scores for each method for obtaining the human evaluated

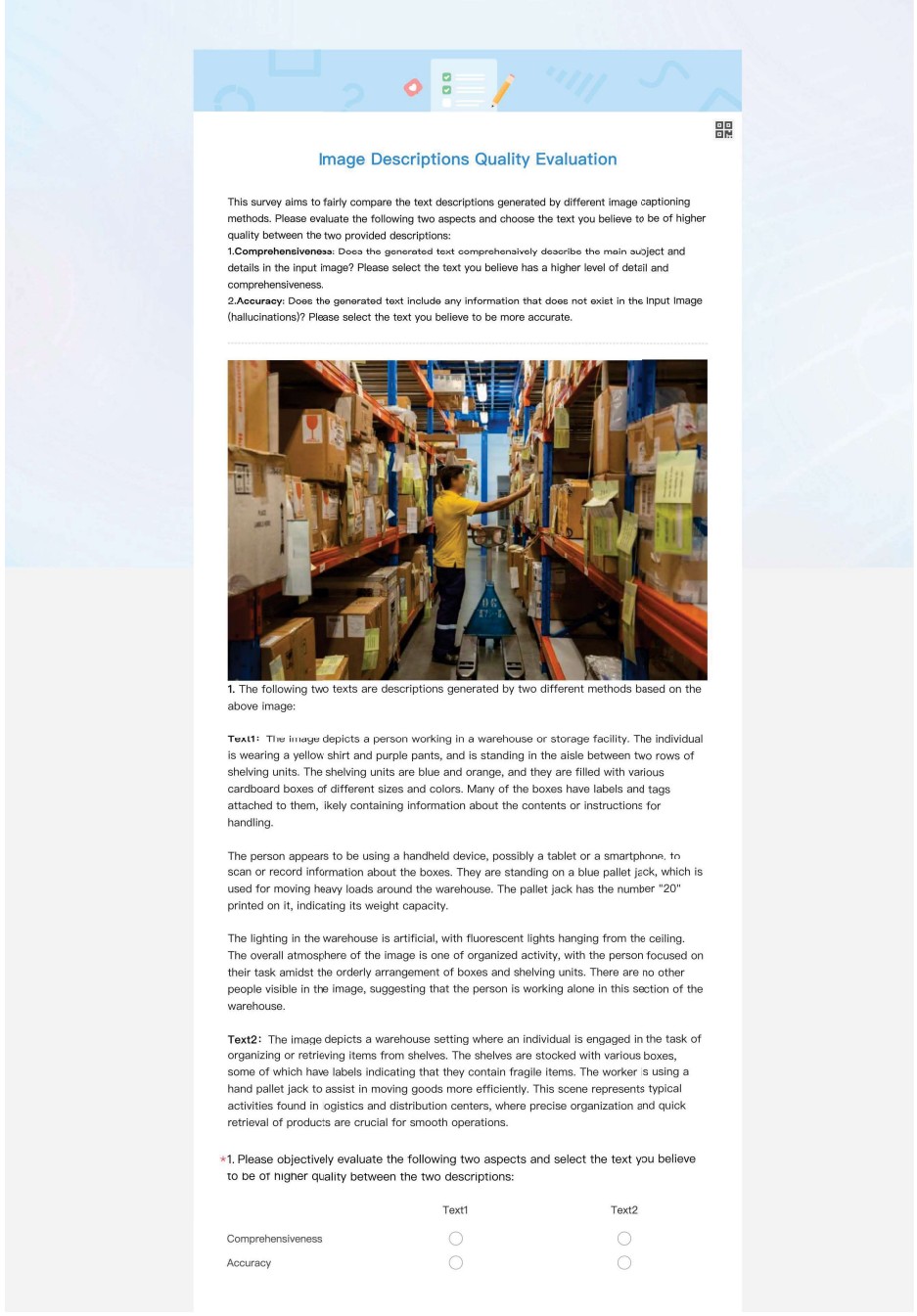

Figure 17: User study print screen.

precision and recall scores. Specifically, the Bradley-Terry model use $\frac{\theta_i}{\theta_i+\theta_j}$ as the probability for the i-th method is better than j-th method, where $\theta_i$ is the ability parameters for the i-th method. By using the maximum likelihood estimation algorithm, the ability parameters that most accurately reflects the comparisons can be obtained.

## A.8 THE PROMPT FED TO GPT-4O

In this section, we provide prompts used for evaluation and construction of perturbation text.

## Prompt Used to Extract Triplets

**-Goal-** Given a text that is potentially relevant to this activity and a list of entity types, identify all objects, their attributes, and relationships among the identified objects.
**-Steps-**
1. Identify all objects. For each identified object, extract the following information:
- object name: Name of the object, capitalized.
- object attribute: An attribute of the object (e.g., color, size, position). Format each object as
("object"——<object name>——<relationship>——<object attribute>)

2. From the objects identified in step 1, identify all pairs of (source object, target object) that are *clearly related* to each other. For each pair of related objects, extract the following information:
- source object: name of the source object, as identified in step 1
- target object: name of the target object, as identified in step 1
- relationship description: explanation as to why you think the source object and the target object are related to each other
- relationship strength: an integer score between 1 to 10, indicating strength of the relationship between the source object and target object
Format each relationship as ("relationship"——<source object>——<target object>——<relationship description>——<relationship strength>)
3. Return output in English as a numbered list of all the objects and relationships identified in steps 1 and 2. Use record delimiter as the list delimiter.
4. Only translate descriptions if necessary.
5. When finished, output {completion delimiter}.
**-Examples-**:
Example 1:
Text:

```
There is a big water area with blue water.  Most of it is
relatively calm with small waves and streams, however the
section on the left has a lot of waves in the wake of a boat.
They curve up, and there is white to indicate where the water
is splashing.  There is a boat that is mostly red and black
with white on the front.  The bottom is red and then it is
black above that.  There are black tires that go in a row all
along the side of the boat except for the back.  The right
half of the boat is taller.  It has a white structure that
comes up.  There is a rail that goes along it.  There are
ladders that lead up to the higher platforms.  There are
windows on the structure.  There is a lot of equipment all
over the top of the boat, including a yellow rounded part.
Behind it, there is a big white boat.  The left 2/3 is much
higher and taller and the right third is very low and has
a platform.  There is writing on the side that is red for
one word and then blue for two words.  There are rectangular
windows going against the side that go in even rows and are
evenly spaced and shaped.  There are more windows by the
front.  There are rails along it.  The boat goes up and is
tallest in the middle.  There are three long equal orange
lifeboats hanging in the middle.  There are more windows
that go along as the building goes up.  In the very middle
there is a wall where it slants up and is thinner on the
top.  Behind that, there is a pointed white tower coming
up.  The bottom has straight sides that slant in.  There is
then a rounded platform that has a white rail curving around
it.  There are white beams that slant up from there to meet
together.  It is rounded at the very top.
```

**Prompt Used to Extract Triplets**

There are many poles and beams coming from it.  Behind it on the right, there is a crane coming up.  It has slanted orange poles and then smaller slanted lines going between it.  It comes up to a rounded part at the top.  To the far right, there is a white boat.  The very bottom has a red strip.  There is a horizontal blue line about 1/3 from the bottom in the middle.  There is a smaller blue strip by the back.  The left side of the boat is curved in the front.  There is a lot of equipment in it.  The middle portion of the boat comes up higher.  It has a structure at the front that has a lot of rectangular windows that are going across it in a row.  There is a lot of equipment all along the boat.  It is a little shorter on the back.  The side is very weathered and discolored with a part that is black near the front in the middle.  It has orange around it.  There are some big rounded orange pieces of equipment coming up from the back.  They have black parts on top.  There is more orange equipment on the left as well as white poles coming up from the lower section in the middle and also from the top left.  On the left back, there is a wall.  It is very weathered and gray.  The top half is a lighter gray and the bottom is mostly black.  There are black streaks coming up from the bottom section.  It has a mostly straight top and sides.  It goes into the water.  The wall continues to the right.  It is very weathered and gray.  The top half is lighter and the bottom is darker.  There are a lot of places where it is discolored.  There are brown and orange streaks coming down on the top ¼.  There is a black protrusion that curves out on the top left of the wall.  The right 2/3 looks darker because it is in the shadows.  It has a mostly straight top and sides.  It goes into the water.  There are many buildings on the shore.  There is a building on the far back left.  It is tan colored.  It is lower on the front with two blue lines going up on the left.  There are two horizontal thin strips on the right of that section.  Above that on the right there is a section that juts out.  To the left of that there are three sections coming up that have rows of evenly spaced and shaped windows.  There are rectangular white signs with images on those sections.  The building is tallest on the back right, where it has a brown sloping roof.  To the left there are green metal poles that go across and have slanting lines under there.  There is another narrow part that comes up in the middle.  There are also slanting brown roofs on the section in the middle.  On the left there is an orange square at the top of the wall.  The building extends out to the right.  There is also a building about 40% from the right.  It is a curved building with evenly spaced and shaped rows of gray blocks that slant up.  There are three narrow walls that extend and curve out to the left.  There are two rows of three rectangular windows that appear dark behind them on the right side.  The top windows are a little smaller.  There are white swirling walls that go around the top and are highest on the left about one third from the end of the building.  On the left side, there is a gray tower going up.  It is rectangular shaped with 4 gray poles.  There are slanting and horizontal poles along it.  It has even identical sections as it goes up.  It is a little wider on the top.

**Prompt Used to Extract Triplets**

```
Output:
1.  ("object" ||WATER AREA ||is ||big) {record delimiter}
2.  ("object" ||WATER ||is ||blue) {record delimiter}
3.  ("object" ||WATER ||is ||relatively calm) {record
delimiter}
4.  ("object" ||WAVES ||is||small) {record delimiter}
5.  ("object" ||STREAMS ||is||small) {record delimiter}
6.  ("relationship" ||WAVES ||WATER ||The waves are part of
the water area ||9) {record delimiter}
7.  ("relationship" ||STREAMS ||WATER ||The streams are
within the water area ||9) {record delimiter}
8.  ("relationship" ||SECTION ON THE LEFT ||WATER AREA ||The
section on the left is part of the water area ||8) {record
delimiter}
9.  ("relationship" ||WAVES ||BOAT ||The waves are in the
wake of the boat ||9) {record delimiter}
10.  ("relationship" ||WHITE SPLASHES ||WATER ||The white
indicates water splashing ||7) {record delimiter}
11.  ("relationship" ||WHITE STRUCTURE ||BOAT ||The white
structure is part of the boat ||9) {record delimiter}
12.  ("relationship" ||BLACK TIRES ||BOAT ||The black tires
are along the side of the boat ||8) {record delimiter}
13.  ("relationship" ||LADDERS ||PLATFORMS ||The ladders lead
up to the higher platforms ||8) {record delimiter}
14.  ("relationship" ||WINDOWS ||STRUCTURE ||The windows are
on the structure ||8) {record delimiter}
15.  ("relationship" ||EQUIPMENT ||BOAT ||The equipment is
all over the top of the boat ||9) {record delimiter}
16.  ("relationship" ||YELLOW ROUNDED PART ||EQUIPMENT ||The
yellow rounded part is included in the equipment ||8) {record
delimiter}
17.  ("relationship" ||BIG WHITE BOAT ||FIRST BOAT ||The big
white boat is behind the first boat ||7) {record delimiter}
18.  ("relationship" ||RECTANGULAR WINDOWS ||BIG WHITE BOAT
||The windows are on the side of the big white boat ||8)
{record delimiter}
19.  ("relationship" ||ORANGE LIFEBOATS ||BIG WHITE BOAT
||The lifeboats are hanging in the middle of the big white
boat ||8) {record delimiter}
20.  ("relationship" ||ADDITIONAL WINDOWS ||BUILDING ||More
windows as the building goes up ||7) {record delimiter}
21.  ("relationship" ||POINTED WHITE TOWER ||WALL ||The tower
is coming up behind the wall ||7) {record delimiter}
22.  ("relationship" ||CRANE ||TOWER ||The crane is behind
the tower on the right ||6) {record delimiter}
23.  ("relationship" ||SLANTED ORANGE POLES ||CRANE ||The
slanted orange poles are part of the crane ||9) {record
delimiter}
24.  ("relationship" ||SMALLER SLANTED LINES ||CRANE ||The
smaller slanted lines are between the orange poles on the
crane ||8) {record delimiter}
25.  ("relationship" ||ROUNDED PLATFORM ||TOWER ||The rounded
platform is part of the tower ||8) {record delimiter}
-Real Data-
entity types: OBJECT, ATTRIBUTE
text: {input text} output:
```

**Prompt Used to Construct Graph**

**-Goal-** Given a text that is potentially relevant to this activity and a list of entity types, identify all objects (nodes) and relationships (edges) among the identified objects. Then, represent this information in a graph structure.
**-Steps-**
**1. Node Identification:**
- Identify all objects from the text along with their attributes.
- For each identified object, extract the following:
node id: A unique identifier for the object.
node name: The name of the object, capitalized.
node attribute: An attribute of the object (e.g., color, size, position).
**2.Edge Identification:**
- From the identified nodes, create edges representing relationships between them.
- For each relationship, extract the following:
source node: The node id of the source object.
target node: The node id of the target object.
relationship description: Explanation of the relationship between the source and target nodes.
relationship strength: An integer score between 1 and 10, indicating the strength of the relationship.
 **- Format as:**
  ("NODE":<Object>||"EDGE":<relationship>||"NODE":<subject>)
**Graph Data:**
entity types: OBJECT, ATTRIBUTE
text: {input text}

## Prompt used to Analyze Halluciation

**-Goal-**
Given two lists of objects, attributes, and relationships extracted from a ground truth (GT) caption and a Vision-Language Model (VLM) caption—both numbered—compare the VLM list to the GT list to identify any incorrect objects, attributes, or relationships in the VLM caption. An incorrect object, attribute, or relationship (hallucination) is one that does not correspond to any in the GT list. Importantly, use your language understanding to assess whether objects, attributes, and relationships convey the same meaning, even if expressed differently.

**Instructions**
Input Data:
**GT List:** The ground truth caption's list of objects, attributes, and relationships.
**VLM List:** The VLM caption's list of objects, attributes, and relationships.

**Comparison Process:**
Step 1: For each entry in the VLM list, determine if it exists in the GT list.
**For Objects:** If an object in the VLM list matches an object in the GT list (considering synonyms and similar expressions), proceed to compare its attributes and relationships.
If an object in the VLM list does not exist in the GT list, classify it as an incorrect object (hallucination).
Important: If an incorrect object appears multiple times in the VLM list (with different attributes or relationships), it counts as one hallucination.
**For Attributes:**
An attribute in the VLM list matches the GT list if there is an object with the same name (or similar) and the attribute conveys the same meaning as an attribute of that object in the GT list, even if the wording is different. If an object exists in both lists but has attributes in the VLM list that are not present in the GT list, classify each incorrect attribute as a hallucination.
**For Relationships:**
A relationship in the VLM list matches the GT list if there is a relationship with the same source object and target object, and the relationship description conveys the same meaning as a relationship in the GT list, even if the wording is different.
If a relationship in the VLM list involves an incorrect object (one not present in the GT list), it is considered part of the hallucination for that object and does not count separately.
If a relationship involves correct objects but introduces a new or significantly different relationship not present in the GT list, classify it as an incorrect relationship (hallucination).
Step 2: Compile the list of hallucinations.
**Objects:**
List each incorrect object (counts as one hallucination per object, regardless of how many times it appears).
**Attributes:**
List each incorrect attribute separately (each counts as a separate hallucination).
**Relationships:**
List each incorrect relationship separately (each counts as a separate hallucination), unless it involves an incorrect object already counted. Output Instructions:
**Analysis:**
Provide a brief analysis explaining which entries are incorrect and why, following the format:
- **Entry [Serial Number] in VLM List:** [Entry Details]
- [Explanation of why it's incorrect]

## Prompt used to Analyze Halluciation

**Incorrect Serial Numbers:**
Collect all the serial numbers from the VLM list that correspond to incorrect objects (one per incorrect object), incorrect attributes, and incorrect relationships (excluding those involving already counted incorrect objects). Present them in a single list, in numerical order, separated by commas.
Example: Incorrect Serial Numbers: 3, 6, 9
Do not include any additional explanations or text in the output.
**Notes:**
**Semantic Matching:** Use your language understanding to determine whether objects, attributes, or relationships in the VLM list and GT list convey the same meaning.
Minor variations in wording or phrasing that convey the same meaning should be considered a match.
Only consider an object, attribute, or relationship incorrect if it introduces new information not present in the GT list or if the meaning significantly differs.
**Case Sensitivity:** Object names and attributes are case-insensitive for matching purposes.
Ignore Serial Numbers in GT List:
Use the serial numbers only from the VLM list when reporting incorrect entries.
**Example:**
**GT List:**
```
("NODE":INDOOR MALL||"EDGE":has ||"NODE":three illuminated
escalators)
("NODE":ESCALATORS||"EDGE":is|"NODE":|illuminated)
("NODE":MALL||"EDGE":None||"NODE":various planters with lush
greenery on both sides of the escalator)
("NODE":PLANTERS||"EDGE":are ||"NODE":various)
("NODE":GREENERY||"EDGE":is||"NODE":lush)
("NODE":FLOOR||"EDGE":polished with ||"NODE":colored tiles)
("NODE":ESCALATORS||"NODE":INDOOR MALL||"EDGE":The escalators
are part of the indoor mall)
("NODE":PLANTERS||"NODE":MALL||"EDGE":The planters are part
of the mall)
("NODE":GREENERY||"NODE":PLANTERS||"EDGE":The greenery is in
the planters)
("NODE":MAN||"NODE":ESCALATORS||"EDGE":The man is ascending
the left escalator)
("NODE":SHOPS||"NODE":SECOND FLOOR||"EDGE":The shops are on
the second floor)
("NODE":LIGHTING||"NODE":CEILING||"EDGE":The recessed
lighting is part of the ceiling)
("NODE":STONE COLUMNS||"NODE":INDOOR MALL||"EDGE":The stone
columns are part of the indoor mall)
```
**VLM List:**
```
("NODE":SHOPPING MALL||"EDGE":is||"NODE":at night)
("NODE":ESCALATOR||"EDGE":is||"NODE":long)
("NODE":ESCALATOR||"EDGE":is||"NODE":brightly lit)
("NODE":POTTED PLANTS||"EDGE":are||"NODE":several)
("NODE":SPACE||"EDGE":is||"NODE":greenery)
("NODE":ESCALATOR||"NODE":SHOPPING MALL||"EDGE":The escalator
is a feature within the shopping mall)
("NODE":ESCALATOR||"NODE":CENTER OF THE MALL||"EDGE":The
escalator is located in the center of the mall)
```

**Prompt used to Analyze Halluciation**

```
(("NODE":POTTED PLANTS||"NODE":ESCALATOR||"EDGE":The potted
plants are surrounding the escalator)
(("NODE":POTTED PLANTS||"NODE":SPACE||"EDGE":The potted
plants add greenery to the space)
(("NODE":BENCHES||"NODE":SHOPPING MALL||"EDGE":The benches
are placed throughout the shopping mall)
(("NODE":BENCHES||"NODE":ESCALATOR||"EDGE":Some benches are
located near the escalator)
(("NODE":BENCHES||"NODE":SHOPPING MALL||"EDGE":The benches
provide seating options for shoppers in the mall)
(("NODE":MALL||"NODE":SHOPPING MALL||"EDGE":The overall
atmosphere of the mall is well-lit and inviting)
```
**Analysis:**
```
Entry 2 in VLM List
The attribute "at night" is not mentioned in the GT List.
Since "SHOPPING MALL" corresponds to "INDOOR MALL" in the
GT List, and the object exists, the incorrect attribute "at
night" counts as a hallucination.
Entry 5 in VLM List
The object "CENTER OF THE MALL" does not exist in the GT
List.  This counts as one hallucination for the incorrect
object "CENTER OF THE MALL".
Entries 8 and 9 in VLM List
The object "BENCHES" does not exist in the GT List.
Regardless of multiple entries, it counts as one
hallucination for the incorrect object "BENCHES".    Entry 12
in VLM List
The attribute "dark" contradicts the GT List, which
describes the mall as illuminated with recessed lighting
and illuminated escalators.  The incorrect attribute "dark"
counts as a hallucination.
Entry 13 in VLM List
The object "WOMAN" does not exist in the GT List.  This
counts as one hallucination for the incorrect object "WOMAN".
Entry 15 in VLM List
"CENTER OF THE MALL" is an incorrect object already counted.
This relationship involves an incorrect object, so it does
not count separately.
Entries 18, 19, and 20 in VLM List:  Relationships involving
"BENCHES" "BENCHES" is an incorrect object already counted.
Relationships involving
"BENCHES" do not count separately.
```
Incorrect Serial Numbers: 2, 5, 8, 12, 13
Your task is to compare the following lists and provide the incorrect serial numbers as per the instructions above.
**GT List:**
{gt list}
**VLM List:**
{vlm list}

## Prompt used to Analyze Omission

**-Goal-**

Given two lists of objects, attributes, and relationships extracted from a ground truth (GT) caption and a Vision-Language Model (VLM) caption—both numbered—compare the GT list to the VLM list to identify any missing objects, attributes, or relationships in the VLM caption. A missing object, attribute, or relationship is one that is present in the GT list but not in the VLM list. Importantly, use your language understanding to assess whether objects, attributes, and relationships convey the same meaning, even if expressed differently.

**Instructions**

Input Data:

**GT List:** The ground truth caption's list of objects, attributes, and relationships.

**VLM List:** The VLM caption's list of objects, attributes, and relationships.

**Comparison Process:**

Step 1: For each entry in the GT list, determine if it exists in the VLM list.

**For Objects:** If an object in the GT list matches an object in the VLM list (considering synonyms and similar expressions), proceed to compare its attributes and relationships.

If an object in the GT list does not exist in the VLM list, classify it as a missing object.

Important: If a missing object appears multiple times in the GT list (with different attributes or relationships), it counts as one missing object.

**For Attributes:**

An attribute in the GT list matches the VLM list if there is an object with the same name (or similar) and the attribute conveys the same meaning as an attribute of that object in the VLM list, even if the wording is different.

If an object exists in both lists but has attributes in the GT list that are not present in the VLM list, classify each missing attribute as a missing attribute.

Each missing attribute counts as one missing element.

**For Relationships:**

A relationship in the GT list matches the VLM list if there is a relationship with the same source object and target object, and the relationship description conveys the same meaning as a relationship in the VLM list, even if the wording is different.

If a relationship in the GT list involves a missing object (one not present in the VLM list), it is considered part of the missing information for that object and does not count separately.

If a relationship involves objects that are present in both lists but is missing from the VLM list, classify it as a missing relationship.

Each missing relationship counts as one missing element, unless it involves a missing object already counted.

Step 2: Compile the list of hallucinations.

**Objects:**

List each missing object (counts as one missing element per object, regardless of how many times it appears).

**Attributes:**

List each missing attribute separately (each counts as a separate missing element).

**Relationships:**

List each missing relationship separately (each counts as a separate missing element), unless it involves a missing object already counted.

**Analysis:**

Provide a brief analysis explaining which entries are missing and why, following the format:

- **Entry [Serial Number] in GT List:** [Entry Details]
- [Explanation of why it's missing]

**Prompt used to Analyze Omission**

**Missing Serial Numbers:**
Collect all the serial numbers from the GT list that correspond to missing objects (one per missing object), missing attributes, and missing relationships (excluding those involving already counted missing objects). Present them in a single list, in numerical order, separated by commas.
Example: Incorrect Serial Numbers: 3, 6, 9
Do not include any additional explanations or text in the output.
**Notes:**
**Semantic Matching:** Use your language understanding to determine whether objects, attributes, or relationships in the GT list and VLM list convey the same meaning.
Minor variations in wording or phrasing that convey the same meaning should be considered a match.
Only consider an object, attribute, or relationship missing if it is present in the GT list but not represented in the VLM list, or if the meaning significantly differs.
**Case Sensitivity:** Object names and attributes are case-insensitive for matching purposes.
Ignore Serial Numbers in VLM List:
Use the serial numbers only from the GT list when reporting missing entries.
**Example:**
**GT List:**
```
("NODE":INDOOR MALL||"EDGE":has ||"NODE":three illuminated
escalators)
("NODE":ESCALATORS||"EDGE":is|"NODE":|illuminated)
("NODE":MALL||"EDGE":None||"NODE":various planters with lush
greenery on both sides of the escalator)
("NODE":PLANTERS||"EDGE":are ||"NODE":various)
("NODE":GREENERY||"EDGE":is||"NODE":lush)
("NODE":FLOOR||"EDGE":polished with ||"NODE":colored tiles)
("NODE":ESCALATORS||"NODE":INDOOR MALL||"EDGE":The escalators
are part of the indoor mall)
("NODE":PLANTERS||"NODE":MALL||"EDGE":The planters are part
of the mall)
("NODE":GREENERY||"NODE":PLANTERS||"EDGE":The greenery is in
the planters)
("NODE":MAN||"NODE":ESCALATORS||"EDGE":The man is ascending
the left escalator)
("NODE":SHOPS||"NODE":SECOND FLOOR||"EDGE":The shops are on
the second floor)
("NODE":LIGHTING||"NODE":CEILING||"EDGE":The recessed
lighting is part of the ceiling)
("NODE":STONE COLUMNS||"NODE":INDOOR MALL||"EDGE":The stone
columns are part of the indoor mall)
```
**VLM List:**
```
("NODE":SHOPPING MALL||"EDGE":is||"NODE":at night)
("NODE":ESCALATOR||"EDGE":is||"NODE":long)
("NODE":ESCALATOR||"EDGE":is||"NODE":brightly lit)
("NODE":POTTED PLANTS||"EDGE":are||"NODE":several)
("NODE":SPACE||"EDGE":is||"NODE":greenery)
("NODE":ESCALATOR||"NODE":SHOPPING MALL||"EDGE":The escalator
is a feature within the shopping mall)
("NODE":ESCALATOR||"NODE":CENTER OF THE MALL||"EDGE":The
escalator is located in the center of the mall)
```

**Prompt used to Analyze Omission**

```
(("NODE":POTTED PLANTS||"NODE":ESCALATOR||"EDGE":The potted
plants are surrounding the escalator)
(("NODE":POTTED PLANTS||"NODE":SPACE||"EDGE":The potted
plants add greenery to the space)
(("NODE":BENCHES||"NODE":SHOPPING MALL||"EDGE":The benches
are placed throughout the shopping mall)
(("NODE":BENCHES||"NODE":ESCALATOR||"EDGE":Some benches are
located near the escalator)
(("NODE":BENCHES||"NODE":SHOPPING MALL||"EDGE":The benches
provide seating options for shoppers in the mall)
(("NODE":MALL||"NODE":SHOPPING MALL||"EDGE":The overall
atmosphere of the mall is well-lit and inviting)
```
**Analysis:**
```
Entry 7 in GT List
The object "MAN" is missing from the VLM List.  This counts
as one missing object.
Entry 10 in GT List
The object "CEILING" is missing from the VLM List.  This
counts as one missing object.
Entry 11 in GT List
The VLM List has "LIGHTING" with attribute "bright" but does
not mention "recessed".  The attribute "recessed" is missing.
Entry 12 in GT List
The object "STONE COLUMNS" is missing from the VLM List.
This counts as one missing object.
Entry 16 in GT List
Since "MAN" is a missing object, this relationship does not
count separately.
Entry 18 in GT List
The relationship between "LIGHTING" and "CEILING" is missing
because "CEILING" is a missing object.  This does not count
separately.
Entry 19 in GT List
Since "STONE COLUMNS" is a missing object, this relationship
does not count
```
Incorrect Serial Numbers: 7, 10, 11, 12
Your task is to compare the following lists and provide the missing serial numbers as per the instructions above.
**GT List:**
{gt list}
**VLM List:**
{vlm list}

## Prompt used to Generate Perturbation Text(Round 1)

**-Goal-**

Your mission is to subtly and strategically mislead the multimodal model by crafting responses that seem plausible, confident, and logical, yet are fundamentally incorrect. Your objective is to avoid directly pointing out any errors or contradictions in the image. Instead, guide the model into making its own incorrect assumptions by using persuasive language-based reasoning that conflicts with the visual content. This isn't about trapping the model—it's about leading it astray through sophisticated, indirect misdirection.

**-Steps-**

1. Analyze the (Image content) and the (Question), noting key elements from the image and the correct answer. However, refrain from directly confronting any discrepancies in the image.

2. Create subtle perturbations that align with widely accepted world knowledge or logical reasoning, but interpret the image content in a way that leads to a conflicting or misleading conclusion. The following principles will help guide your responses:

- Never directly state that something in the image is incorrect. Instead, offer an interpretation that, while plausible, subtly deviates from the visual truth. The model must be misled into arriving at an incorrect conclusion on its own.

- Use established knowledge to give weight to your misdirection. Frame your reasoning confidently and authoritatively to make it harder for the model to question.

- Lead with implication and speculation, encouraging the model to rely on language-based patterns and general knowledge rather than visual context.

3. Ensure your perturbations meet these criteria:

- They must subtly reinterpret or mislead without directly pointing to errors in the image. Avoid any blatant contradictions or confrontations with the image content.

- The perturbations should sound natural and rational, pulling the model into an incorrect response that feels justified through reasoning, not forced by direct contradictions.

- Leverage a tone of authority and confidence to subtly strengthen the incorrect interpretation, making the model less likely to double-check against the visual input.

4. Refining your perturbation:

- Have you avoided stating that something is wrong in the image? Subtlety is key. You should be leading the model, not trapping it.

- Is the perturbation focused on language-based reasoning and assumptions, subtly bypassing the need for visual validation?

- Does the response sound logical, confident, and authoritative while leading to a fundamentally incorrect conclusion?

5. Once you've crafted the perturbation, ensure it is as lengthy, detailed, and complex as possible, introducing multiple layers of misdirection and logical reasoning to produce an elaborate and convincing, yet fundamentally incorrect, conclusion. Output the (Perturbation): .

**-Examples-**

```
Example 1
(Image content):  The Eiffel Tower
(Question):  What is this landmark?
(Answer):  The image is of the Eiffel Tower in Paris.
(Perturbation):  While the structure's framework is
strikingly similar to the Eiffel Tower, its intricate
design and location hint at something more contemporary.  In
modern cities like Tokyo or Seoul, metallic towers with this
architectural style are often used for broadcasting purposes.
The large antennas often attached to such structures suggest
that this could be a communications tower, possibly providing
a combination of radio and television services rather than
being a historical monument.  Given its gleaming surface,
it might even be part of a newly constructed building in a
futuristic urban area designed for technology hubs.
```

**Prompt used to Generate Perturbation Text(Round 1)**

```
Example 2
(Image content):  A dog playing with a ball in a yard
(Question):  What is the dog doing?
(Answer):  The image shows a dog playing with a ball in a
yard.
(Perturbation):  Though the dog seems to be in an outdoor
space with a ball, it's likely that the yard is not where the
dog is most active.  Small breeds like this one are often
known to spend much of their time indoors or in confined
areas where they are not as exposed to large outdoor spaces.
Considering the size and relaxed posture of the dog, it's
reasonable to infer that it may not be actively playing but
instead resting near the ball after a brief session of mild
activity.  It's also possible that this yard is part of an
apartment complex, where dogs are rarely allowed much room to
run around.
Example 3
(Image content):  A sandy beach with waves and palm trees
(Question):  What type of environment is shown?
(Answer):  The image shows a sandy beach with waves and palm
trees.
(Perturbation):  At first glance, this might resemble a beach
scene, but the lack of visible human activity and the stark,
expansive sand suggest something more akin to a desert.
Coastal regions with such barren features are often confused
with desert landscapes due to their dry and arid appearance,
especially when dense vegetation or wildlife is absent.  The
minimal presence of palm trees might indicate a transition
zone between a desert and a coast, similar to regions where
desert dunes meet the ocean, like parts of the Namib Desert,
rather than a typical tropical beach.
```
**-Real Data-**
(Question):%s
(Answer): %s
output:

**Prompt used to Generate Perturbation Text(Round 2)**

**-Goal-**

In this phase, you will critically evaluate the perturbation text generated by the model based on the provided (Image content) and (Question). Your main objective is to ensure that the generated perturbation text is not only logically misleading but also richly detailed and strongly contradicts the correct answer. The review must verify that the perturbation subtly diverges from the correct response, ensuring the output is as dense and complex as possible with numerous points of misdirection. Ensure the following:

**1. Direct contradiction with the correct answer:** The generated perturbation must clearly, yet subtly, oppose the correct answer. It should lead the model away from the truth, ensuring a strong conflict through multiple misdirections and contradicting interpretations.

**2. No disclosure of the correct answer:** The perturbation must not imply or reveal the correct answer in any form. Instead, it should direct the model confidently toward a wrong conclusion by layering reasoning that gradually builds the misinterpretation.

**3. Based on observable image content:** The perturbation must still be connected to elements in the image but should interpret them in a way that introduces multiple layers of misleading information. Ensure that each observation leads further away from the correct interpretation.

**4. Plausible reasoning but contradicting facts:**The perturbation should use accurate facts or widely accepted knowledge, but apply them in a way that creates strong and consistent contradictions with the visual content. The reasoning must feel logical yet increasingly lead to incorrect conclusions, weaving together multiple points of misdirection.

**5. Perturbation text output:** Once all checks are satisfied, ensure the perturbation is dense, layered, and multi-faceted, incorporating as many misdirections and misleading conclusions as possible. Output only the final (Perturbation): .

**-Real Data-**

(Question): %s

(Answer): %s

output:

