# OpenReview forum: "PerturboLLaVA: Reducing Multimodal Hallucinations with Perturbative Visual Training"
_ICLR.cc/2025/Conference — ICLR 2025 Spotlight_

### Official Review · Reviewer_3ZD9 · 2024-10-23

**Soundness:** 3
**Presentation:** 3
**Contribution:** 3
**Rating:** 8
**Confidence:** 3

**Summary:**

The paper has two main contributions: (1) Introducing an evaluation pipeline for dense image captioning, HalFscore, that measures the quality of dense caption more holistically by constructing a language graph and computing the precision and recall. (2) Introducing an adversarial augmentation method during training that reduces the reliance on language prior, which reduces hallucination.

**Strengths:**

1. This paper addresses a significant gap in the current evaluation of MLLMs
2. The augmentation method is intuitive, and shows nice improvements in the benchmarks checked
3. The authors conduct extensive experiments to demonstrate the effectiveness of the augmentation method

**Weaknesses:**

1. Regarding the human survey, it is unclear to me who the human raters are (the authors, Amazon Mechanical Turk Workers, etc.)
2. Since all the experiments are done on LLaVA-1.5, it is unclear what the effect of the perturbation method on stronger models

**Questions:**

If the annotators are from Amazon Mechanical Turk, how were they recruited?
If the annotators reported in the paper are the authors, could you also add a comparison with external annotators?

---

> ### Author Response · Authors · 2024-11-17
>
> **Q1: About human survey**
>
> Thank you for your attention to this part of our experiments. The survey was conducted anonymously among junior students from the computer science department. Screenshots of the questionnaire has been included in the appendix A.7.
>
> **Q2: The effect of the perturbation method on stronger model**
>
> | **Model**                        | **LLM Backbone** | **Vision Encoder** | **Training Data** | **MMB_EN ↑** | **Precision ↑** | **Recall ↑** | **F1-Score ↑** | **CHAIR_s ↓** | **CHAIR_i ↓** | **HallBench ↑** | **CCBench ↑** | **SEED ↑** |
> | -------------------------------- | ---------------- | ------------------ | ----------------- | ------------ | --------------- | ------------ | -------------- | ------------- | ------------- | --------------- | ------------- | ---------- |
> | **Stronger Model**               | Qwen2-7B         | SigLIP-400M        | LLaVA1.5          | 70.4         | 59.8            | 48.4         | 53.5           | 48.1          | 14.2          | 52.1            | 46.1          | 65.6       |
> | **Stronger Model + Our Methods** | -                | -                  | -                 | 70.2         | 60.9            | 49.7         | 54.0           | 30.9          | 7.4           | 52.4            | 47.0          | 65.6       |
>
> To further validate our method’s effectiveness on stronger models, we conducted experiments using the same network architecture as MiniCPM-V-2.6 but trained on LLaVA 1.5’s data. From the results in the table above, it is evident that our method effectively reduces hallucinations and also enhances model performance on more advanced models. Note that we didn't further fine-tune state-of-the-art (SOTA) multimodal models. The reason is that we found directly fine-tuned InternVL2 on the open-source LLaVA 665k dataset led to a significant drop in performance due to the quality gap of the training data. We believe that conducting experiments under such a setting would make it challenging to derive solid conclusions. Therefore, we opted to reproduce the network structure of an open-source SOTA model while still using the publicly available LLaVA 1.5 dataset, showcasing the capability of our approach on a stronger model.

---

> > ### Comment · Reviewer_3ZD9 · 2024-11-22
> >
> > I thank the authors for the additional experiment and clarification, I raise my score accordingly.

---

> > > ### Comment · Reviewer_3ZD9 · 2024-11-23
> > > **Additional relevant related work**
> > >
> > > Seems like [1] is also a relevant related work, they also suggest a method for dense image captioning evaluation based on precision and recall.
> > >
> > > [1] Yanuka et al 2024. https://arxiv.org/abs/2411.09018

---

> ### Author Response · Authors · 2024-11-23
>
> We are delighted to see that the concerns raised by the reviewer have been successfully addressed. We sincerely appreciate your great efforts in reviewing this paper.
>
> Thank you for bringing up [1] and we have reviewed it carefully. [1] primarily focuses on the known knowledge of multimodal models and proposes the DNLI evaluation metric for dense captions and the KnowAda data augmentation method. Although [1] also employs precision and recall as metrics, [1] places greater emphasis on evaluating the consistency of responses to dense captions directly generated by the multimodal model and questions reformulated by LLMs. These evaluations are used to define the Descriptiveness metric and the Contradiction metric.  While our metric directly compares the captions output by the multimodal model with ground-truth captions to determine precision and recall metrics, which reflect the model's hallucination level and descriptiveness of the image.
>
> We believe that [1]  is valuable in its consideration of which responses in the dense captions generated by a multimodal model stem from lack of capability (random guessing) and which responses arise from misinterpretations despite recognizing the visual objects accurately. However, our concern with [1] lies in the use of thresholds based on multimodal model responses to LLM-generated questions to distinguish between known and unknown knowledge, which may not be sufficiently accurate. Additionally, we believe [1] could further explore why multimodal models output unknown knowledge instead of choosing not to respond.
>
> [1] Yanuka et al 2024. https://arxiv.org/abs/2411.09018

---

### Official Review · Reviewer_bhZ8 · 2024-11-03

**Soundness:** 3
**Presentation:** 3
**Contribution:** 3
**Rating:** 8
**Confidence:** 5

**Summary:**

This paper aims to address the issue of hallucination in current multimodal large language models applied to dense image captioning tasks. The authors argue that there is no appropriate metric for evaluating the hallucinations in this context. To address this gap, they introduce HalFscore, a metric based on a language graph that assesses both the accuracy and completeness of dense captions. Additionally, they propose the incorporation of perturbed text to alleviate the over-reliance of language-prior issue, thereby mitigating the hallucination issue. Experimental results on several general multimodal benchmarks demonstrate the effectiveness of their approach.

**Strengths:**

1. The proposed PerturboLLaVA is simple, interesting and effective. By modifying the original training data without employing additional training techniques or supplementary data, this method effectively addresses the hallucination issue when compared to the baseline LLaVA-1.5 (Table 2 & Table 3).

**Weaknesses:**

1. Defination of dense captioning task is unclear:
As noted in Lines 012 and 038, this paper focuses on addressing the hallucination problem in dense captioning tasks. The authors reference [1] and [2], both of them are related to dense captioning tasks. However, the definitions of dense captioning tasks in [1] and [2] differ from those presented in this paper. Additionally, this paper does not conduct standard evaluations for these tasks as outlined in [1] and [2]. It remains unclear whether this discrepancy constitutes a citation issue, and the authors should provide clarification on this matter.

[1] DenseCap: Fully Convolutional Localization Networks for Dense Captioning, 2016.
[2] Context and attribute grounded dense captioning, 2019.

2. No comparison between HalFscore and existing methods:
The authors solely rely on user studies to justify the effectiveness of the proposed metric. However, they do not provide a comparison with existing metrics. One commonly used baseline is to using gpt-score as evaluation metric (e.g., [3])

[3] LLaVA-RLHF: Aligning Large Multimodal Models with Factually Augmented RLHF, 2023.


3. Additional training cost compare with existing methods.

The methods need to re-train the whole model and cost more training time (the training samples are longer).

4. No evaluation on stronger models.

The paper conducts experiments solely on LLaVA-1.5, which is publicly available as of 2023. More recent and stronger models, such as LLaVA-Next, Intern VL-1.5, and LLaVA-UHD, are not evaluated. It is recommended that the authors include at least one of these recent models to demonstrate the effectiveness of their methods.

**Questions:**

Please refer to weakness

---

> ### Author Response · Authors · 2024-11-17
>
> **Q1: Definition of dense caption model**
>
> We are very grateful to the reviewers for raising this issue. We apologize for confusing you with the definition here.The dense caption task we studied about the Multi-Modal Large Language Model is different from these two papers. Our task can be understood as providing detailed natural language descriptions for images, which is also mentioned in previous MLLM works[3][4][5].So we don't need to do the detection and localization task mentioned in paper [1] and [2], we have updated the paper's citation to make it clarified.
>
> [1] DenseCap: Fully Convolutional Localization Networks for Dense Captioning, 2016.
>
> [2] Context and attribute grounded dense captioning, 2019.
>
> [3]Visual Instruction Tuning,2023
>
> [4]ShareGPT4V: Improving Large Multi-Modal Models with Better Captions,2023
>
> [5]InternVL: Scaling up Vision Foundation Models and Aligning for Generic Visual-Linguistic Tasks,2024
>
> **Q2: Comparision with gpt-based metric**
>
> As detailed in our paper, we have already compared against a GPT-based scoring metric, MMHalBench, which is reflected in the user study (see Table 6 and lines 516–518 in the main text). MMHalBench, introduced in the RLAIF-V paper, is a GPT-based scoring metric that provides an overall score ranging from 0 to 6 for model responses. Our superior performance in human preference evaluations on hallucinations supports the validity of our metric.
>
> Moreover, compared to score-based evaluation methods, our metric provides significant advantages. It allows for fine-grained assessment of various types of hallucinations in generated captions and facilitates detailed analysis, whereas score-based metrics are limited to providing a single aggregated score (e.g., 0–10).
>
> **Q3: Additional training cost**
>
> | **Cost of Training**          | **Average Memory Cost (GB)** | **Training Time Cost (min)** |
> | ----------------------------- | ---------------------------- | ---------------------------- |
> | **Baseline**                  | 62.3                         | 264                          |
> | **PerturboLLaVA**             | 63.8                         | 281                          |
> | **Additional Overhead Ratio** | 2.5%                         | 6.4%                         |
>
> Unlike RLHF-V, RLAIF-V, or LLaVA-RLHF, which introduce an additional training stage after the SFT phase, our approach enhances the model's capability directly during the SFT phase by incorporating perturbation training through the insertion of perturbation texts. The training overhead in our method is minimal, stemming solely from the increased token length caused by the augmented data during the SFT phase. Furthermore, we have quantified the average GPU memory usage and training time overhead introduced by incorporating perturbation text. As shown in the table below, the additional training overhead is negligible. Compared to methods such as LLaVA-RLHF and RLAIF-V, which require extra training stages or reward modeling, our approach demonstrates clear advantages. A detailed analysis of this comparison has also been provided in the appendix A.3 for further reference.

---

> ### Author Response · Authors · 2024-11-17
>
> **Q4: Evaluation of the effect of preturbative method on stronger model**
>
> Thanks for your good suggestion. To further validate our method’s effectiveness on stronger models, we conducted experiments using the same network architecture as MiniCPM-V-2.6 which uses siglip-400M and qwen2-7B but trained on LLaVA 1.5’s data. From the results in the table below, it is evident that our method effectively reduces hallucinations and also enhances model performance on more advanced models.
>
> | **Model**                        | **LLM Backbone** | **Vision Encoder** | **Training Data** | **MMB_EN ↑** | **Precision ↑** | **Recall ↑** | **F1-Score ↑** | **CHAIR_s ↓** | **CHAIR_i ↓** | **HallBench ↑** | **CCBench ↑** | **SEED ↑** |
> | -------------------------------- | ---------------- | ------------------ | ----------------- | ------------ | --------------- | ------------ | -------------- | ------------- | ------------- | --------------- | ------------- | ---------- |
> | **Stronger Model**               | Qwen2-7B         | SigLIP-400M        | LLaVA1.5          | 70.4         | 59.8            | 48.4         | 53.5           | 48.1          | 14.2          | 52.1            | 46.1          | 65.6       |
> | **Stronger Model + Our Methods** | -                | -                  | -                 | 70.2         | 60.9            | 49.7         | 54.0           | 30.9          | 7.4           | 52.4            | 47.0          | 65.6       |
>
> Note that we didn't further fine-tune state-of-the-art (SOTA) multimodal models. The reason is that we found directly fine-tuned InternVL2 on the open-source LLaVA 665k dataset led to a significant drop in performance due to the quality gap of the training data. We believe that conducting experiments under such a setting would make it challenging to derive solid conclusions. Therefore, we opted to reproduce the network structure of an open-source SOTA model while still using the publicly available LLaVA 1.5 dataset, showcasing the capability of our approach on a stronger model.
>
> Meanwhile, our reproduced stronger model has already significantly outperformed LLaVA-next-Vicuna-7B and comparable with LLaVA-Next-Llama3, making it a sufficiently robust baseline to demonstrate the effectiveness of our method.
>
> Furthermore, from a methodological perspective, applying our approach directly during the SFT phase aligns more closely with the original intent of our method.
>
> | Type                      | MMB_EN | MMMU_VAL | HallusionBench | SEEDImage | CCBench | ScienceQA_Test |
> | ------------------------- | ------ | -------- | -------------- | --------- | ------- | -------------- |
> | LLaVA-Next-Llama3         | 69.8   | 43.1     | 33.1           | 72.5      | 32.7    | 73.1           |
> | LLaVA-Next-Vicuna-7B      | 63     | 37.6     | 27.6           | 69.6      | 24.3    | 70.3           |
> | Reproduced Stronger Model | 70.4   | 40.5     | 52.1           | 65.6      | 46.1    | 73.6           |

---

> ### Comment · Reviewer_bhZ8 · 2024-11-18
> **Further Concerns on Performance**
>
> As the authors address most of my concerns, I increase my rating. However, my main concern remains - I don't see significant improvements in stronger models when using the proposed method. This could be a limitation for future follow-up work. I hope the author can further address my concern.

---

> ### Author Response · Authors · 2024-11-22
>
> Thank you for your timely feedback and for recognizing the additional experiments and conclusions we presented. We sincerely appreciate your interest in our work and aim to address your concerns as follows.
>
>
> | **Model**                          | **LLM Backbone** | **Vision Encoder** | **Training Data** | **Precision ↑** | **Recall ↑** | **F1-Score ↑** | **CHAIR_s ↓** | **CHAIR_i ↓** | **HallBench ↑** | **CCBench ↑** | **SEED ↑** |
> | ---------------------------------- | ---------------- | ------------------ | ----------------- | --------------- | ------------ | -------------- | ------------- | ------------- | --------------- | ------------- | ---------- |
> | **Stronger Model**                 | Qwen2-7B         | SigLIP-400M        | LLaVA1.5          | 59.8            | 48.4         | 53.5           | 48.1          | 14.2          | 52.1            | 46.1          | 65.6       |
> | **Stronger Model + Our Methods**   | -                | -                  | -                 | 60.9            | 49.7         | 54.0           | 30.9          | 7.4           | 52.4            | 47.0          | 65.6       |
> | **LLaVA1.5**                       | Vicuna-7B        | CLIP ViT-L/14      | LLaVA1.5          | 53.3            | 45.8         | 49.2           | 54.2          | 15.0          | 46.9            | 29.4          | 65.3       |
> | **LLaVA1.5 + Our Methods**         | -                | -                  | -                 | 59.5            | 46.5         | 52.2           | 36.1          | 10.4          | 47.5            | 30.6          | 65.6       |
> | **LLaVA1.5 + Our Methods + OPERA** | -                | -                  | -                 | 60.2            | 47.0         | 52.8           | 33.1          | 10.1          | 47.6            | 31.0          | 65.6       |
>
> From the perspective of performance improvements, we observe that our method achieves consistent gains on both LLAVA 1.5 and stronger models. While the HalFscore Precision metric shows less significant improvement on the stronger model, we believe this is likely due to the strong baseline performance, which inherently limits further gains. To meet your requirements for stronger models, we utilized an internally pre-trained version of siglip instead of the publicly available HuggingFace version. This siglip was trained on additional high-quality data and supports dynamic resolutions ranging from 378 to 2k, allowing for improved extraction of visual information. By the way, we are also actively working on training a sota multimodal model. The training of the stronger model also directly utilized image resolutions up to 2k, which surpasses the 336 resolution limit of LLAVA 1.5 imposed by CLIP. Meanwhile, considering the performance of other sota models in terms of precision, we can observe that for strong models, precision is a relatively challenging metric to improve significantly. Therefore, we believe that the improvements brought by our method on a stronger baseline are effective and reasonable. Moreover, we achieved a significant improvement in the hallucination metric Chairs for dense captioning. This also highlights the effectiveness of our method on stronger models.
>
> | Model                        | Size | Precision ↑ | Recall ↑ | Fscore ↑ |
> | ---------------------------- | ---- | ----------- | -------- | -------- |
> | Ovis1.6-Gemma2               | 9B   | 61.5        | 50.3     | 55.4     |
> | Qwen2-VL                     | 7B   | 60.8        | 50.0     | 54.9     |
> | LLaVa-onevision              | 7B   | 61.3        | 48.3     | 54.1     |
> | Stronger Model + Our Methods | 7B   | 60.9        | 49.7     | 54.0     |
> | InternVL2                    | 8B   | 60.6        | 48.6     | 53.9     |
> | Stronger Model               | 7B   | 59.8        | 48.4     | 53.5     |
>
> Here, we would like to delve deeper into the topic of improving the performance of multimodal models. Based on technical reports of sota models and our own training experience, it is evident that the network structures, backbones, and training strategies used by these models are quite similar. The outstanding performance of SOTA models primarily stems from differences in the scale and quality of the training data used. For instance, LLaVA-OneVision curated over a hundred high-quality open-source datasets and sampled the highest-quality data for training. InternVL and DeepSeek-VL rely heavily on a significant proportion of in-house high-quality data. If we aim to further enhance the performance of stronger models, the key lies in utilizing high-quality training data and applying our perturbation training strategy on such data.

---

> ### Author Response · Authors · 2024-11-22
>
> | **Type**            | **HalFscore↑** |            |            | **Object HalBench ↓** |             | **HalBench ↑** | **MMB ↑** | **SEED ↑** | **CCBench ↑** |
> | ------------------- | -------------- | ---------- | ---------- | --------------------- | ----------- | -------------- | --------- | ---------- | ------------- |
> |                     | **Precision**  | **Recall** | **Fscore** | **CHAIR_s**           | **CHAIR_i** |                |           |            |               |
> | **LLaVA1.5**        | 53.3           | 45.8       | 49.2       | 54.2                  | 15.0        | 46.9           | 67.3      | 65.3       | 29.4          |
> | **LLaVA1.5 + 40k**  | 54.3           | 45.9       | 49.8       | 52.3                  | 14.8        | 46.9           | 67.3      | 65.4       | 30.2          |
> | **LLaVA1.5 + 80k**  | 56.1           | 46.2       | 50.6       | 47.9                  | 13.2        | 49.2           | 68.0      | 65.1       | 31.6          |
> | **LLaVA1.5 + 120k** | 58.7           | 46.6       | 52.0       | 42.2                  | 10.8        | 49.1           | 68.7      | 64.8       | 29.2          |
> | **LLaVA1.5 + 160k** | 59.5           | 46.5       | 52.2       | 36.1                  | 10.4        | 47.5           | 68.9      | 65.6       | 30.6          |
>
> Our method indeed has the ability to scale up, effectively reducing hallucination in stronger models by increasing the scale of perturbed data. To substantiate this, we have provided supplementary experiments based on discussions with reviewer RVA9. These experiments clearly show that as the scale of perturbed data increases, our method continues to reduce multimodal model hallucination. Considering the consistent performance improvements our method has demonstrated on both LLAVA 1.5 and stronger models, we are confident that increasing the scale of perturbed data will further enhance the performance of stronger models. However, the perturbed data we use for stronger models already includes all 160k VQA-related data from LLAVA 1.5. Further increasing the scale of perturbed data would require incorporating additional training datasets, generating corresponding perturbed data, and retraining the baseline. Due to limitations in time and computational resources, we regretfully could not conduct these additional experiments. However, based on the current experimental results, our conclusion that the performance of models on hallucination metrics improves with the scale of perturbed data remains valid.
>
> From the perspective of addressing hallucination, most mainstream methods focus on training additional reward models after the SFT stage or improving performance during the inference stage. In contrast, our approach reduces the model's reliance on language priors and enhances its visual comprehension during the SFT stage itself. Our method provides an additional solution pathway for addressing hallucination, one that can be seamlessly combined with other present methods for further improvements. The experiments we conducted in the paper, integrating our approach with the Opera strategy, also validate this point.
>
> In conclusion, we believe that the over-reliance of multimodal models on language priors is a fundamental challenge contributing to hallucination. This issue stems from the inherent characteristics of current training paradigms, where models initialized with language models tend to prioritize language priors over visual information. Effectively addressing this challenge is, in our view, a key step toward reducing hallucination of VLM. We are hopeful that our method, along with the insights shared, will provide a useful perspective and inspire future researchers to further explore and develop more comprehensive solutions.

---

> ### Comment · Reviewer_bhZ8 · 2024-11-22
> **Good Paper**
>
> It is appreciated that the authors actively participate in the discussion.
>
> I believe the authors have addressed my concerns. I have increased my rating to 8 and hope some of discussions can be included in the final version of the paper. Good paper!

---

> ### Author Response · Authors · 2024-11-22
>
> We are delighted to see that the concerns raised by the reviewer have been successfully addressed. We sincerely appreciate your great efforts in reviewing this paper. Your constructive advice and valuable comments really help improve our paper. We will include some discussions in the final version of the paper. Once again, we would like to express our heartfelt thanks for your valuable time and insightful feedback.

---

### Official Review · Reviewer_vMbR · 2024-11-04

**Soundness:** 3
**Presentation:** 3
**Contribution:** 2
**Rating:** 5
**Confidence:** 4

**Summary:**

This paper proposes an evaluation metric called HalFscore to measure the fine-grained image captioning quality in terms of object, attribute, and relation hallucinations. The authors also propose a method to reduce hallucinations by injecting generated perturbation text into training prompts and show that it helps reduce the hallucination compared to several baselines.

**Strengths:**

1. The proposed hallucination metric HalFscore is straightforward but helpful in detecting fine-grained hallucination in terms of objects, attributes, and relations.
2. Based on perturbation text, the proposed training framework PerturboLLaVA forces the model to balance the image content and language prior, avoiding generating hallucinated text in fine-grained captions.

**Weaknesses:**

Both HalFscore and PertuboLLaVA are pretty straightforward. PerturboLLaVA is a simple prompting technique to revise the text prompts for model finetuning. The results are mixed, and there are still performance gaps compared to recent models based on advanced language models like Qwen2 and LLama3.

**Questions:**

Questions:
Can you provide more details on the inconsistent results of RLAI-F in Table 3 between Object HalBench (CHAIRs and CHAIRi) and HalBench?
Comments:
Lines 430, 463: The results in "Table 2"... --> Should be "Table 3"

---

> ### Author Response · Authors · 2024-11-17
>
> **W1: HalFscore and PerturboLLaVA are pretty straightforward.**
>
> As highlighted by other reviewers , we respectfully believe that being straightforward is not a weakness of our method; rather, we consider it a strength. Solving complex problems in a simple, fundamental, and intuitive manner is an effective and valuable approach.
>
> Furthermore, we sincerely hope that the insights and contributions underlying our approach will not be overlooked. Detecting hallucinations for all categories in dense captions at a fine-grained level is far from trivial. We demonstrate that the concept graph data structure is particularly effective for this task. Similarly, mitigating the overshadowing effect of linguistic priors on multimodal models is a subtle and challenging issue, as the influence is implicit and difficult to quantify. Our perturbative approach offers an elegant and intuitive solution to this problem.
>
> **W2: Perturbative training on LLaVA 1.5 does not surpass the performance of current** **SOTA** **open-source VLMs.**
>
> - **Baseline** **Gap**: We would like to clarify that the foundational performance of LLaVA 1.5 is significantly lower than that of current SOTA open-source models. These SOTA models achieve substantial improvements through advancements in backbone architectures, training data quality, and training strategies. It should be mentioned that SOTA models benefit from access to massive, high-quality datasets which are extremely time-consuming and expensive—for example, LLaVA-OneVision curated and cleaned over 100 premium open-source datasets, while InternVL leverages a large proportion of in-house high-quality data. Both the quantity and quality of data in these SOTA models far surpass those available to LLaVA 1.5. What's more, unlike LLaVA 1.5, which only relies on Vicuna+CLIP, SOTA multimodal models leverage stronger LLM backbones such as Qwen2 and LLaMA 3, as well as advanced vision encoders like SigLIP and InternViT, resulting in significant performance gains. From a training strategy perspective, LLaVA 1.5 fine-tunes only the projector and LLM, whereas SOTA models generally include vision encoder fine-tuning, yielding substantial performance improvements. Furthermore, the computational cost of training LLaVA 1.5 (6.5 hours on a single H800 GPU) is significantly lower than that of SOTA models, which can require 100 to 1,000 times the resources. Given this significant baseline gap, it is reasonable that our perturbative method does not outperform SOTA open-source models.
> - **Effectiveness of Our Method**: Nevertheless, our method demonstrates clear improvements over the LLaVA 1.5 baseline. Additionally, our approach can be combined with advanced decoding strategies for further enhancements, as acknowledged by reviewers. To further validate our method’s effectiveness on stronger models, we conducted experiments using the same network architecture as Qwen2VL but trained on LLaVA 1.5’s data. From the results in the table below, it is evident that our method effectively reduces hallucinations and also enhances model performance on more advanced models
>
>
> | **Model**                  | **LLM Backbone** | **Vision Encoder**  | **Training Data** | **MMB_EN ↑** | **Precision ↑** | **Recall ↑** | **F1-Score ↑** | **CHAIR_s ↓** | **CHAIR_i ↓** | **HallBench ↑** | **CCBench ↑** | **SEED ↑** |
> |----------------------------|------------------|---------------------|-------------------|--------------|-----------------|--------------|----------------|---------------|---------------|-----------------|---------------|------------|
> | **Stronger Model**         | Qwen2-7B        | SigLIP-400M         | LLaVA1.5          | 70.4         | 59.8           | 48.4        | 53.5          | 48.1          | 14.2          | 52.1            | 46.1          | 65.6       |
> | **Stronger Model + Our Methods** | -              | -                   | -                 | 70.2         | 60.9            | 49.7         | 54.0           | 30.9          | 7.4           | 52.4            | 47.0          | 65.6       |
> | **LLaVA1.5**               | Vicuna-7B       | CLIP ViT-L/14       | LLaVA1.5          | 67.3         | 53.3            | 45.8         | 49.2           | 54.2          | 15.0          | 46.9            | 29.4          | 65.3       |
> | **LLaVA1.5 + Our Methods** | -               | -                   | -                 | 68.9         | 59.5            | 46.5         | 52.2           | 36.1          | 10.4          | 47.5            | 30.6          | 65.6       |
> | **LLaVA1.5 + Our Methods + Opera** | -           | -                   | -                 | 68.9         | 60.2            | 47.0         | 52.8           | 33.1          | 10.1          | 47.6            | 31.0          | 65.6       |

---

> ### Author Response · Authors · 2024-11-17
>
> **Q1: Evaluation of RLAIF-V**
>
> We sincerely appreciate your thorough review and insightful observation. Upon carefully re-examining our code, we identified an issue in the prompt design for RLAIF-V when evaluating HallusionBench. Specifically, we did not enforce the requirement for the model to "directly output Yes or No," which resulted in mismatched answer inputs. We are deeply grateful for your careful reading and keen feedback. In our revised version, we have addressed this issue along with the other points mentioned in your comments. Thank you once again for your detailed and thoughtful suggestions!
>
> **Q2:"Table 2" -> "Table 3"**
>
> Once again, thank you very much for your careful reading. We have corrected this issue in the updated PDF version

---

> ### Author Response · Authors · 2024-11-26
>
> Dear Reviewer vMbR,
>
> We greatly appreciate the time you took to review our paper. With the discussion phase nearing its conclusion, we kindly ask for your feedback on whether we have sufficiently addressed your main concerns.
>
> We remain fully committed to clarifying or elaborating on any points that might still require further attention. Please don’t hesitate to let us know if additional explanations would be helpful.
>
> Thank you once again for your time and constructive input!

---

### Official Review · Reviewer_nju1 · 2024-11-04

**Soundness:** 3
**Presentation:** 3
**Contribution:** 3
**Rating:** 8
**Confidence:** 4

**Summary:**

This paper studies hallucination problem in MLLM, especially on dense captioning task. The authors first propose a new metric that can capture both precision recall based on scene graph parsed by a GPT model. The author then propose a training method PerturboLLava which can reduce the model’s reliance on language prior so that the model can focus more on the visual inputs thus reducing hallucination. The authors show the effectiveness of the proposed method on a selected 1k images from DCI dataset.

**Strengths:**

Nice ablation on different version of training templates as well as different relevance levels of perturbation.
The proposed method can be combined with other decoding-based methods to further improve performance.
The authors introduce a new well motivated metric and show that it correlates more strongly with human judgment.
The method is conceptually simple and effective, requiring no additional inference cost. It not only enhances dense captioning but also benefits general multimodal capabilities. Although the authors primarily focus on the dense captioning task, the method is not designed with no specificity on captioning but generally aims to help the model focus on image input over language priors.

**Weaknesses:**

The proposed metric is very similar to SPICE(SPICE: Semantic Propositional Image Caption Evaluation) however the authors do not mention this related work. The main difference is SPICE is using a traditional parser but this paper uses GPT. I hope the authors can add the comparison against SPICE.

I think figure 1 is not super accurate. The authors do use extra data (although augmented from existing data), and also the longer training examples will lead to extra training cost as well.

Having Section 4.2 is good, but I have some concern. I do think equation 5-10 holds, but I think the explanation following the equations is problematic. “During the training of multimodal models, we can assume that the world knowledge embedded in the language model remains unchanged, so p(xk|xp<k) is a perturbation term that cannot be optimized. Therefore, in the training process of multimodal models, our perturbation training method guides the model to optimize towards p(xk|x−p<k,I), transforming it into a fully multimodal model that is unaffected by language priors and relies entirely on image information to answer questions.”
- The statement here holds no matter with or without perturbation.
- My own explanation is, for cross entropy loss, if the model already predicts well, the gradient will be low so the model wouldn’t pay much attention to this token. Without the perturbation, the prior term can already predicts well, so the gradient is low. With the perturbation, the prior term is perturbed to be low, so the gradient is high for the unprior term.
- Essentially originally, for a caption, the tokens that can be inferred from language prior will have less effect on parameter gradients whereas now the gradient is kind of balanced across tokens.

**Questions:**

- In table4,5 the authors show different variants of perturbative visual training. I am wondering if the authors try combining these different versions.
- The training and inference has some discrepancy where training has the perturbation as input but inference does not. Why this kind of discrepancy is not an issue?
- The metric highly rely on the scene graph representation. What are the limitations of this representation?
- Is there analysis on how good the gpt4 model perform perturbation task?

---

> ### Author Response · Authors · 2024-11-17
>
> **W1: Comparison against SPICE**
>
> We are very grateful to the reviewer for pointing out the work of SPICE. Please forgive us for neglecting this important work during previous research. We will add references to SPICE and similar methods in the final version. Actually, we have tried some different methods when exploring how to divide the scene graph, and finally we decided to use GPT as our scene parser for the following reasons：
>
> 1. In the dense captioning task of MLLM, the length of the caption is often very long, it is hard for traditional scene parsers to catch the contextual information that is far apart, but thanks to the pre-training of a large amount of data, GPT is excel at doing such chllenging work.
> 2. GPT has strong  ability to do in-context learning, so we can show examples for GPT to make the output in the format we desired.
> 3. The performance of GPT can be further enhanced with well-designed prompts as shown in our appendix.
>
> Here we provide a simple comparison between GPT and a SPICE like method FACTUAL[1](which scene parser is stronger than that in SPICE) to show the superiority of GPT, it may be a little unfair because the FACTUAL scene parser cannot accept prompts as complex as GPT，but the results do illustrate the issue to some extent.
>
> **Origin caption:**
>
> Three children are playing in the snow in front of a log cabin. One child is wearing a blue jacket, another is wearing a pink jacket, and the third child is wearing an orange jacket. They are all wearing hats, with one child wearing an orange hat and the other two wearing blue hats.There is a car parked in front of the cabin.
>
> **FACTUAL result:**
>
> ```Plain
> ( child, wear, hat )
> ( child, wear, hat )
> ( child, play in, snow )
> ( child, wear, hat )
> ( child, wear, hat )
> ( car, park in front of, cabin )
> ( child, play in, snow )
> ( hat, is, orange )
> ( child, wear, hat )
> ( child, wear, hat )
> ( child, play in, snow )
> ( child, wear, hat )
> ( child, play in, snow )
> ( hat, is, orange )
> ```
>
> **GPT** **result:**
>
> ```Python
> 1.("NODE":CHILDREN||"EDGE":are||"NODE":three)
> 2.("NODE":CHILDREN||"EDGE":on||"NODE":SNOW)
> 3.("NODE":LOG CABIN||"EDGE":behind||"NODE":BOY)
> 4.("NODE":CHILDREN||"NODE":SNOW||"EDGE":The children are playing in the snow)
> 5.("NODE":CHILD||"NODE":BLUE JACKET||"EDGE":One child is wearing a blue jacket)
> 6.("NODE":CHILD||"NODE":PINK JACKET||"EDGE":One child is wearing a pink jacket)
> 7.("NODE":CHILD||"NODE":ORANGE JACKET||"EDGE":One child is wearing an orange jacket)
> 8.("NODE":CHILDREN||"NODE":HATS||"EDGE":All children are wearing hats)
> 9.("NODE":CHILD||"NODE":ORANGE HAT||"EDGE":One child is wearing an orange hat)
> 10.("NODE":CHILDREN||"NODE":BLUE HATS||"EDGE":Two children are wearing blue hats)
> 11.("NODE":CAR||"NODE":LOG CABIN||"EDGE":The car is parked in front of the log cabin)
> ```
>
> Through the comparison of qualitative results, it can be seen that traditional scene parsers cannot receive complex prompts, lack a deep understanding of contextual content, and can't output ideal formats as GPT does.
>
> [1] Li Z, Chai Y, Zhuo T Y, et al. FACTUAL: A Benchmark for Faithful and Consistent Textual Scene Graph Parsing[C] Findings of the Association for Computational Linguistics: ACL 2023. 2023: 6377-6390.

---

> ### Author Response · Authors · 2024-11-17
>
> **W2: About the explanation of Table**
>
> Thank you for your careful review of the table.
>
> - Regarding the use of additional data, although we only augmented the existing data, the introduction of an additional perturbation text generation process does incur extra overhead. Therefore, when comparing the data-related costs of different methods, we still consider that we have introduced additional data overhead to ensure a fair comparison. We realize that the current description could be more precise. We have revised this part of the text accordingly in the manuscript. Thank you for your thorough review and for bringing this to our attention.
>
> | **Cost of Training**          | **Average Memory Cost (GB)** | **Training Time Cost (min)** |
> | ----------------------------- | ---------------------------- | ---------------------------- |
> | **Baseline**                  | 62.3                         | 264                          |
> | **PerturboLLaVA**             | 63.8                         | 281                          |
> | **Additional Overhead Ratio** | 2.5%                         | 6.4%                         |
>
> - As for computational overhead, we sincerely appreciate your feedback. Our original intent was to claim that we do not require an additional training phase after the SFT stage. However, it is true that adding perturbation text increases the number of tokens, which can also be a concern. We have revised the table in this section. Furthermore, we have quantified the average GPU memory usage and training time overhead introduced by incorporating perturbation text. As shown in the table below, the additional training overhead is negligible. Compared to methods such as LLaVA-RLHF and RLAIF-V, which require extra training stages or reward modeling, our approach demonstrates clear advantages. A detailed analysis of this comparison has also been provided in the appendix for further reference.
>
> **W3: About the explanation of equation 5-10**
>
> We greatly appreciate your attention to this part of the content. We strongly agree with your interpretation of the role of perturbed text from the perspective of gradient optimization. In fact, our **Figure 5** on loss feedback is intended to reflect this viewpoint: for cases misled by perturbed text, the gradient becomes larger, driving the model toward focusing more on the image and being less affected by the language prior.
>
> We can delve into the multimodal model's training process. **Equation 10**, $p(x_k \mid x^p_{<k}, I) $ , can be understood as the behavior of the multimodal model leveraging intrinsic language priors and world knowledge to answer questions, while $p(x_k \mid x^{-p}_{<k}, I)$ represents the model's ability to utilize the image to answer questions.
>
> When we use a language model to initialize the multimodal model, the multimodal model can only rely on language capability to answer questions. On multimodal training data, relying on language priors often leads to incorrect token predictions, i.e., $p(x_k \mid x^p_{<k}, I)$ becomes too small. In such cases, the gradient drives the model to improve $p(x_k \mid x^{-p}_{<k}, I)$, optimizing the model's visual capability.
>
> If we further introduce perturbed text during training, $p(x_k \mid x^p_{<k}, I)$ decreases further, meaning that the probability of predicting correctly based on language ability reduces. Consequently, a larger gradient is generated to optimize and improve $p(x_k \mid x^{-p}_{<k}, I)$. Due to space constraints in the paper, we may not have fully explained the role of perturbed text, and we sincerely apologize for that. Based on your suggestions, we have supplemented the theoretical explanation for the perturbation strategy: by comparing the differences in cross-entropy gradients before and after perturbation, we found that they can be decomposed into three terms, corresponding to pre-perturbation, post-perturbation DPO reward, and the log probability difference in language model priors before and after perturbation. Therefore, we believe our perturbation method effectively enhances learning efficiency against language model priors. Specific derivations will be added in the appendix.
>
> Additionally, from the perspective of the training data distribution, there is an interesting issue raised by **Reviewer RVA9**. When the multimodal training data entirely lies within the distribution of language priors, the model can sufficiently answer questions and make correct predictions by relying solely on language ability. In this case, $p(x_k \mid x^p_{<k}, I)$ becomes sufficiently large, leading to insufficient optimization of $p(x_k \mid x^{-p}_{<k}, I)$. Conversely, if we introduce distributions that contradict world knowledge and go beyond the boundaries of the language model's knowledge—for example, constructing image distributions such as a kitten playing with a dinosaur along with corresponding data—this could theoretically improve the model's ability to interpret visual inputs.

---

> ### Author Response · Authors · 2024-11-17
>
> **Q1: Combining different variants of perturbative visual training**
>
>
> | **Type**            | **HalFscore↑** |            |            | **Object HalBench↓** |               | **HalBench↑** | **MMB↑** | **SEED↑** | **CCBench↑** |
> | ------------------------- | -------------- | ---------- | ---------- | --------------------- | ------------- | -------------- | --------- | ---------- | ------------- |
> |                     | **Precision**  | **Recall** | **Fscore** | **CHAIR_s**           | **CHAIR_i** |                |           |            |               |
> | **LLaVA1.5**        | 53.3           | 45.8       | 49.2       | 54.2                  | 15.0          | 46.9           | 67.3      | 65.3       | 29.4          |
> | **Version1**                | 59.5           | 46.5       | 52.2       | 36.1                 | 10.4        | 47.5          | 68.9      | 65.6      | 30.6         |
> | **50%Version1+50%Version3** | 59.5           | 46.4       | 52.1       | 30.9                 | 10.4        | 48.2          | 66.9      | 64.8      | 29.0         |
> | **Version3** | 59.7           | 46.1       | 52.0         | 32.3                 | 10.6        | 49.6          | 66.5      | 64.7      | 28.8         |
>
> We are very grateful for the proposal given by the reviewer. Our purpose in designing v1v2v3 is to discuss the impact of perturbation degree. The guiding information of v1v2v3 is from strong to weak, and the degree of perturbation is from weak to strong. v4 setting is to further explore the impact of random perturbation text on training results. We provide an additional experiment to explore the combination of different versions(0.5v1+0.5v3),the result show that the combination perfomance is between v1 and v3, as we expected, idicating that a strong perturbation during training can lead to better performance.
>
> **Q2: The discrepancy between training and inference**
>
> Thank you for pointing this out. It is true that there is a discrepancy between training and inference in this method. We have considered this issue during the design process, but we believe it does not pose a significant problem for the following reasons:
>
> - To prevent the multimodal model from overfitting to the perturbation text in the training paradigm, we designed a diverse set of guiding prompts. As shown in Table 4, for Version 1, we created approximately 100 hint prompts for both before and after the perturbation prompt, and one is randomly selected for each instance. This approach alleviates the risk of the model overfitting to the training paradigm.
> - Not all training data contains perturbation text. In our experiments, augmented data accounts for only about 25% of the total training data. Therefore, the gap between training and inference is not as significant as it might appear.
> - This point involves a deeper discussion. Autoregressive models based on transformers inherently exhibit a discrepancy between training and inference. During parallelization training, each token prediction is conditioned on absolutely correct ground-truth tokens from preceding positions. However, during inference, predictions can be influenced by errors in previously generated tokens. In a sense, our perturbation text simulates this phenomenon during training, helping to mitigate the gap between training and inference.
>
> **Q3: Potential Limitation of scene representation**
>
> Our limitation lies in that hallucination detection based on scene representation may be overly objective, potentially leading to a gap with human preferences. With the powerful image annotation and understanding capabilities of GPT-4o, we construct an accurate and detailed concept graph. While this allows for an objective and equal-weight analysis of each element when comparing and analyzing hallucinations, it does not fully capture human preferences. Humans tend to prioritize the description of main elements in an image over finer details. For instance, in an image of a boy playing with various toys, missing the boy is far more critical than missing one of the toys. However, our scene graph representation currently treats these cases equivalently. Aligning graph-based objective evaluation methods with human evaluation preferences is a direction we aim to explore further. Specifically, we are considering assigning different weights to graph nodes to better reflect human preferences. We will add the limitation analysis in the updated version.

---

> ### Author Response · Authors · 2024-11-17
>
> **Q4: Analysis on how good the gpt4 model perform perturbation task**
>
> | **Type**                       | **HalFscore↑** |            |            | **Object HalBench↓** |             | **HalBench↑** | **MMB↑** | **SEED↑** | **CCBench↑** |
> | ------------------------------ | -------------- | ---------- | ---------- | -------------------- | ----------- | ------------- | --------- | --------- | ------------ |
> |                                | **Precision**  | **Recall** | **Fscore** | **CHAIR_s**          | **CHAIR_i** |               |           |           |              |
> | **LLaVA1.5**                   | 53.3           | 45.8       | 49.2       | 54.2                 | 15.0        | 46.9          | 67.3      | 65.3      | 29.4         |
> | **LLaVA1.5 + 80k(Qwen2-vl)** | 54.8           | 45.9       | 49.9       | 53.3                 | 14.2        | 47.1          | 67.5      | 65.2      | 30.6         |
> | **LLaVA1.5 + 80k(gpt-o)**      | 56.1           | 46.2       | 50.6       | 47.9                 | 13.2        | 49.2          | 68.0      | 65.1      | 31.6         |
>
> We added an experiment that uses the state-of-the-art multimodal model Qwen2-VL instead of gpt-4o to construct perturbation text. Due to time and computational constraints, we constructed only 80k augmented data samples using Qwen2-vl. The results show that using SOTA VLMs to construct perturbation text is also effective, though it does not perform as well as gpt-4o.

---

> > ### Comment · Reviewer_nju1 · 2024-11-30
> >
> > Thank you for the authors' rebuttal. The rebuttal well addressed my questions. I remain my recommendation of acceptance of this paper.

---

### Official Review · Reviewer_RVA9 · 2024-11-05

**Soundness:** 4
**Presentation:** 4
**Contribution:** 3
**Rating:** 8
**Confidence:** 4

**Summary:**

This paper pointed out the hallucination issue in MLLM, where pretrained linguistic knowledge overshadow the visual information. Therefore, this paper proposes a training strategy by augment the text into the one that conflict with the visual content but still aligns with general knowledge. To quantitize the hallucination in fine-grained fashion, it proposes HalFscore (I actually lol at this wordplay), which uses GPT4-o to generate the two scene graph from the two captions and then use GPT4-o to compare them.

**Strengths:**

1.	This paper proposes a new metric to evaluate fine-grained hallucination while also validating the metric by analyzing the correlation with human
2.	This paper proposes a training method that is more fundamental than contrastive decoding.
3.	The paper also provides persuasive experiment results indicating that the hallucination is reduced while preserving the original general VL understanding capability.

**Weaknesses:**

1.	Use GPT-4o as an additional source for perturbation. However, using GPT-o is always expensive on large-scale datasets. I would like to study the impact of the quality of the adversarial language  that is augmented by open-sourced VLMs such as InternVL/QwenVL

**Questions:**

I would like to study the effect of the size of such adversarially augmented data. Would more of such data be more helpful?

I would like to discuss more deeply about such robustness interrupt. Basically you want to construct the data that is unlikely to be generated purely using language bias, thus able to alleviate the overshadowing of the language knowledge. Can we directly construct this training dataset by asking GPT-o to generate these kinds of questions and answers, which directly achieve the paper's goal, instead of purely doing such perturbing over the given QA data? I am concerned actually this perturbing is decreasing the normal VL understanding capability, but because it reduces the hallucination, the general knowledge evaluation (MMbench) is still good

---

> ### Author Response · Authors · 2024-11-17
>
> **Q1: The effect of size of adversarially augmented data**
>
> | **Type**            | **HalFscore↑** |            |            | **Object HalBench ↓** |             | **HalBench ↑** | **MMB ↑** | **SEED ↑** | **CCBench ↑** |
> | ------------------- | -------------- | ---------- | ---------- | --------------------- | ----------- | -------------- | --------- | ---------- | ------------- |
> |                     | **Precision**  | **Recall** | **Fscore** | **CHAIR_s**           | **CHAIR_i** |                |           |            |               |
> | **LLaVA1.5**        | 53.3           | 45.8       | 49.2       | 54.2                  | 15.0        | 46.9           | 67.3      | 65.3       | 29.4          |
> | **LLaVA1.5 + 40k**  | 54.3           | 45.9       | 49.8       | 52.3                  | 14.8        | 46.9           | 67.3      | 65.4       | 30.2          |
> | **LLaVA1.5 + 80k**  | 56.1           | 46.2       | 50.6       | 47.9                  | 13.2        | 49.2           | 68.0      | 65.1       | 31.6          |
> | **LLaVA1.5 + 120k** | 58.7           | 46.6       | 52.0       | 42.2                  | 10.8        | 49.1           | 68.7      | 64.8       | 29.2          |
> | **LLaVA1.5 + 160k** | 59.5           | 46.5       | 52.2       | 36.1                  | 10.4        | 47.5           | 68.9      | 65.6       | 30.6          |
>
> We have supplemented our work with experiments using adversarially augmented data at different scales. Initially, we constructed 160k augmented data samples; now, we have added experiments using only 40k, 80k, and 120k samples. The results, as shown above, demonstrate that progressively increasing the amount of perturbed text leads to an improvement in hallucination mitigation. Additionally, the model's performance on MMB shows steady improvement; however, the metrics on SEEDImage and CCBench are less consistent.
>
> **Q2: Using sota open-sourced VLMs instead of gpt-o to generate perturbative text**
>
>
> | **Type**            | **HalFscore↑** |            |            | **Object HalBench↓** |               | **HalBench↑** | **MMB↑** | **SEED↑** | **CCBench↑** |
> | ------------------------- | -------------- | ---------- | ---------- | --------------------- | ------------- | -------------- | --------- | ---------- | ------------- |
> |                     | **Precision**  | **Recall** | **Fscore** | **CHAIR_s**           | **CHAIR_i** |                |           |            |               |
> | **LLaVA1.5**        | 53.3           | 45.8       | 49.2       | 54.2                  | 15.0          | 46.9           | 67.3      | 65.3       | 29.4          |
> | **LLaVA1.5 + 80k(Qwen2-vl)** | 54.8         | 45.9     | 49.9     | 53.3               | 14.2       | 47.1       | 67.5   | 65.2      | 30.6         |
> | **LLaVA1.5 + 80k(gpt-o)** | 56.1           | 46.2       | 50.6       | 47.9                  | 13.2          | 49.2           | 68.0      | 65.1       | 31.6          |
>
> We added an experiment that uses the state-of-the-art multimodal model Qwen2-VL instead of gpt-4o to construct perturbation text. Due to time and computational constraints, we constructed only 80k augmented data samples using Qwen2-VL. The results show that using SOTA VLMs to construct perturbation text is also effective, though it does not perform as well as gpt-4o.
>
> **Q3: Deeper Discussion about different ways of robustness interrupt.**
>
> We appreciate and agree with your perspective. To address the issue of language knowledge overshadowing, we think directly constructing data that intentionally contradicts real-world knowledge helps. For example, by adjusting image distributions, we can create scenes that conflict with established world knowledge—such as a kitten playing with a dinosaur. We believe that this type of data, which lies outside the bounds of conventional world knowledge of llm, can effectively reduce the multimodal model's over-reliance on language priors and improve its visual understanding capabilities as well. We plan to further investigate and validate this approach in future work.

---

### Author Response · Authors · 2024-11-17

## General Response:

We sincerely thank all the reviewers for taking their time to carefully read through our paper and provide valuable comments.

### Contrubution:

We sincerely thank reviewers RVA9, nju1, vMbR, bhZ8, 3ZD9 for well recognizing the contribution of our method:

1. The proposed method is simple and effective.(Reviewer nju1, vMbR, bhZ8,3ZD9), more fundamental than contrastive decoding (Reviewer RVA9), and can be combined with other decoding-based methods to further improve performance.(Reviewer nju1).
2. The authors introduce a new well motivated metric and show that it correlates more strongly with human judgment (Reviewer RVA9,nju1).
3. This paper addresses a significant gap in the current evaluation of MLLMs(3ZD9).

### Promising and Persuasive Results:

- All reviewers agree that our proposed method effectively reduces the hallucination issue in fine-grained image captions (Reviewer RVA9,nju1,vMbR, bhZ8,3ZD9) while preserving the original general VL understanding capability (Reviewer RVA9,nju1,3ZD9).
- Our paper provides extensive and persuasive experiment results(Reviewer RVA9,3ZD9), and nice ablations on different version of training templates as well as different relevance levels of perturbation(Reviewer nju1).

### Summary of changes

We have revised our submission and summarized our updates as follows:

- Problem of table description mentioned by Reviewer nju1 is fixed.
- We addressed the numerical issues of RLAIF-V pointed out by Reviewer vMbR.
- Citation related to dense image captioning  in the introduction mentioned by Reviewer bhZ8 has been updated.
- We have added quantitative results on the additional training overhead and included them in the appendix A.3.
- We have added the interface screenshot of the user study and included them in the appendix A.7.

---

### Meta-Review · Area_Chair_qxbT · 2024-12-19

**Metareview:**

This paper studies the hallucination in MLLM. The paper specifically studies the image dense captioning task and points out the root cause of the hallucination is the model biased toward relying on the language prior rather than the image information. To measure the hallucination, the paper proposed a HalFScore which built language graphs on groundtruth caption and the visual LM generated caption. The discrepancy b/w those two graphs are measured as the HalFScore. The paper also proposed a data augmentation technique to improve fighting the hallucination. Concretely, the paper proposed to add a misleading prompt based on language prior to the MLLM training. Although the training time slightly increased, the inference time remained the same and the hallucination reduced.

Strength:
1. The task is quite important and the paper is easy to follow
2. The metrics (HalFScore) is well motivated and has a good alignment with human ratings.
3. The proposed approach achieves good performance.

Weakness:
1. This work might rely on GPT-4o (SoTA MLLM) to generate the misleading prompt for data augmentation.

Given the overwhelmingly support for this paper, I would recommend accept.

**Additional Comments On Reviewer Discussion:**

The reviewers argue:
1. missing related work (SPICE)
2. the performance of using other MLLM instead of GPT-4o for creating the misleading prompts.
3. what if the MLLM is not LLAVA but the other stronger model?

The author provides responses to all of those comments. The additional results justify the effectiveness of the proposed approach.

Noted that one reviewer (vMbR) didn't respond to the author's rebuttal. After reading the review, I think this is a low quality review. Therefore, this review is ignored in the final decision.

---

### Decision · Program_Chairs · 2025-01-22

Accept (Spotlight)